# TERMINAL FLOWER 1-FD complex target genes and competition with FLOWERING LOCUS T

Yang Zhu [1], Samantha Klasfeld[1], Cheol Woong Jeong[1,3,5], Run Jin[1], Koji Goto[2], Nobutoshi Yamaguchi [1,4,5] & Doris Wagner [1✉]

Plants monitor seasonal cues to optimize reproductive success by tuning onset of reproduction and inflorescence architecture. TERMINAL FLOWER 1 (TFL1) and FLOWERING LOCUS T (FT) and their orthologs antagonistically regulate these life history traits, yet their mechanism of action, antagonism and targets remain poorly understood. Here, we show that TFL1 is recruited to thousands of loci by the bZIP transcription factor FD. We identify the master regulator of floral fate, *LEAFY* (*LFY*) as a target under dual opposite regulation by TFL1 and FT and uncover a pivotal role of FT in promoting flower fate via *LFY* upregulation. We provide evidence that the antagonism between FT and TFL1 relies on competition for chromatin-bound FD at shared target loci. Direct TFL1-FD regulated target genes identify this complex as a hub for repressing both master regulators of reproductive development and endogenous signalling pathways. Our data provide mechanistic insight into how TFL1-FD sculpt inflorescence architecture, a trait important for reproductive success, plant architecture and yield.

---

[1] Department of Biology, University of Pennsylvania, 415S. University Ave, Philadelphia, PA 19104, USA. [2] Research Institute for Biological Sciences, Okayaka Prefecture, 7549-1, Kibichuoh-cho, Kaga-gun, Okayama 716-1241, Japan. [3] Present address: LG Economic Research Institute, LG Twin tower, Seoul 07336, Korea. [4] Present address: Division of Biological Science, Graduate School of Science and Technology, Nara Institute of Science and Technology, 8916-5 Takayama, Ikoma, Nara 630-0192, Japan. [5] These authors contributed equally: Cheol Woong Jeong, Nobutoshi Yamaguchi. ✉email: wagnerdo@sas.upenn.edu

Of particular importance for reproductive success of flowering plants is optimal timing of onset of reproductive development and of the transition from branch to floral fate in the inflorescence in response to seasonal cues[1–5]. For example, in plants that flower only once, like *Arabidopsis* and most crops, an early switch to flower formation allows rapid completion of the life-cycle and is beneficial in a short growing season[5–7]. At the same time early onset of flower formation reduces seed set and yield since flowers form in lieu of branches, which support production of more flowers per plant[5–8]. By contrast, delaying flower formation increases branching and total flower number, but prolongs time to seed set[5–8].

Key regulators of seasonal control of onset of reproductive development and of the switch from branch to floral fate in primordia of the inflorescence are members of the phosphatidylethanolamine-binding protein (PEBP) family of proteins[5,6,9,10]. Among these, FT promotes onset of the reproductive phase and flower formation (determinacy), while TFL1 promotes vegetative development and branch fate (indeterminacy)[9,11–13]. *Arabidopsis* flowers in the spring and FT accumulates when the daylength exceeds a critical threshold, while TFL1 is present in both short-day and long-day conditions[2,3,14].

FT and TFL1 are small mobile proteins, which have been implicated in transcriptional regulation but do not have DNA-binding domains[14–18]. Biochemical and genetic studies showed that FT physically interacts with the bZIP transcription factor FD via 14-3-3 proteins and similar interactions have recently been described for TFL1[19–22]. Indeed, despite their antagonistic roles, TFL1 and FT are distinguished by only a small number of non-conservative amino acid changes[11,12,23,24]. FT can be converted into TFL1 and vice versa by a single amino acid substitution and such mutations have been selected for during crop domestication[23–26]. Accumulating evidence suggests that FT acts as a transcriptional co-activator, while TFL1 may either prevent FT activity or act as a co-repressor[23,27]. However, non-nuclear roles have also been described for both TFL1 and FT[28,29].

A key unanswered question is how the florigens modulate plant form—what are the downstream processes they set in motion and what is molecular basis for their antagonism? Here we show that TFL1 is recruited to target loci by the bZIP transcription factor FD. We identify the master regulator of floral fate, LEAFY, as a target under dual opposite regulation by TFL1 and FT and uncover a prominent role for FT in LFY upregulation. We find that the antagonism between TFL1 and FT relies on competition for access to chromatin bound FD at the *LFY* locus and other shared targets. Finally, we identify hundreds of TFL1–FD regulated genes linking this complex not only to repression of master regulators of floral fate, but also of diverse endogenous signalling pathways. The combined data reveals how TFL1 and FT tune inflorescence architecture in response to seasonal cues by altering transcriptional programs that direct primordium fate in the inflorescence.

## Result
**TFL1 is recruited to thousands of loci by the bZIP transcription factor FD**. Mechanistic insight into TFL1 activity has been hampered by low protein abundance. To overcome this limitation and to test the role of TFL1 in the nucleus, we first generated a biologically active, genomic GFP-tagged version of TFL1 (gTFL1-GFP *tfl1-1*) (Supplementary Fig. 1a–c) and identified a developmental stage and tissue where TFL1 accumulates. TFL1 protein strongly accumulated in branch meristems in the axils of cauline leaves in 42-day-old short-day grown plants just prior to the switch to flower formation (Fig. 1a). To conduct TFL1 chromatin immunoprecipitation followed by sequencing (ChIP-seq), we next

isolated shoot apices at this stage for anti-GFP immunoprecipitation. Because TFL1 is present in very few cells and binds chromatin indirectly, we combined eight individual ChIP reactions per replicate to enhance detection. We conducted FD ChIP-seq in analogous fashion using a published, biologically active, genomic fusion protein (gFD-GUS *fd-1*)[20] (Supplementary Fig. 1d). This approach yielded high-quality ChIP-seq data in both cases (Supplementary Figs. 2 and 3a).

In total, we identified 3308 and 4422 significant TFL1 and FD peaks (MACS2 summit qval $\leq 10^{-10}$), respectively (Fig. 1b). The TFL1 peaks significantly overlapped with the FD peaks (72% overlap, $p$ val $< 10^{-300}$, hypergeometric test; Fig. 1b–d). De novo motif analysis of ChIP peak summits identified the G-box *cis* motif, a known FD-binding site[30], as most significantly enriched ($p$ val $< 10^{-470}$) and frequently present (>84%) under TFL1 bound and TFL1/FD co-bound peaks (Fig. 1e and Supplementary Fig. 2). To test whether TFL1 chromatin occupancy is dependent on the presence of FD, we also performed TFL1 ChIP-seq in the *fd-1* null mutant. TFL1 chromatin occupancy was strongly reduced in *fd-1* (Fig. 1c, d). Our data point to a prominent nuclear role for TFL1 and show that FD recruits TFL1 to the chromatin of target loci.

Annotating FD and TFL1 peaks to genes identified 2699 joint TFL1 and FD targets. Gene Ontology (GO) term enrichment analysis implicates these targets in abiotic and endogenous stimulus response and reproductive development (Supplementary Table 1). TFL1 and FD peaks were present at loci that promote onset of the reproductive phase in response to inductive photoperiod[2,3,31] like *GIGANTEA* (*GI*), *CONSTANS* (*CO*), and *SUPPRESSOR OF CONSTANS 1* (*SOC1*) and at loci that promote floral fate[31,32] such as *LFY*, *APETALA1* (*AP1*), and *FRUITFULL* (*FUL*) (Fig. 1f). Identification of these TFL1 and FD co-bound targets fits with the known biological role of TFL1 as a suppressor of onset of reproduction and of flower fate and the proposed molecular function of TFL1 in opposing gene activation[11–13,27].

**LEAFY is under dual opposite regulation by TFL1/FD and FT/FD**. We selected the *LEAFY* (*LFY*) gene, which encodes a master regulator of flower fate[33,34], to further probe the molecular mechanism of action of TFL1. While TFL1 promotes branch fate, LFY promotes flower fate in primordia (Supplementary Fig. 4a–f)[13,33–35]. Using independent biological replicates, we confirmed FD-mediated TFL1 binding to *LFY* by ChIP-qPCR (Supplementary Fig. 4g, h). To test whether *LFY* expression is rapidly repressed by the TFL1–FD complex, we generated transgenic plants expressing a steroid inducible version of TFL1 (TFL1^ER; Supplementary Fig. 5). A single steroid treatment reduced *LFY* levels by 50% after 4 h (Supplementary Fig. 4i). The combined data suggest that the TFL1–FD complex directly represses *LFY*.

To better understand TFL1 recruitment to the *LFY* locus, we identified the genomic region sufficient and the *cis* motifs necessary for TFL1 association with the *LFY* locus. TFL1 and FD peak summits located to the second exon of *LFY* (Fig. 1f and Supplementary Fig. 4g, h) and *LFY* reporters that lacked the second exon were not repressed in response to TFL1 overexpression (Supplementary Fig. 6a, b). Exonic transcription factor-binding sites, although rare, are found in both animals and plants, and frequently link to developmental regulation[36,37]. To test whether LFY exon 2 (e2) alone is sufficient to recruit TFL1–FD, we transformed gTFL1-GFP *tfl1-1* plants with a T-DNA containing only *LFY* e2. We detected strong TFL1 recruitment to the introduced copy of e2, using primer sets that specifically amplify the transgene borne exon (Fig. 2a). Next we identified three putative bZIP-binding sites in the second exon of *LFY*; These include an evolutionarily conserved G-box and two

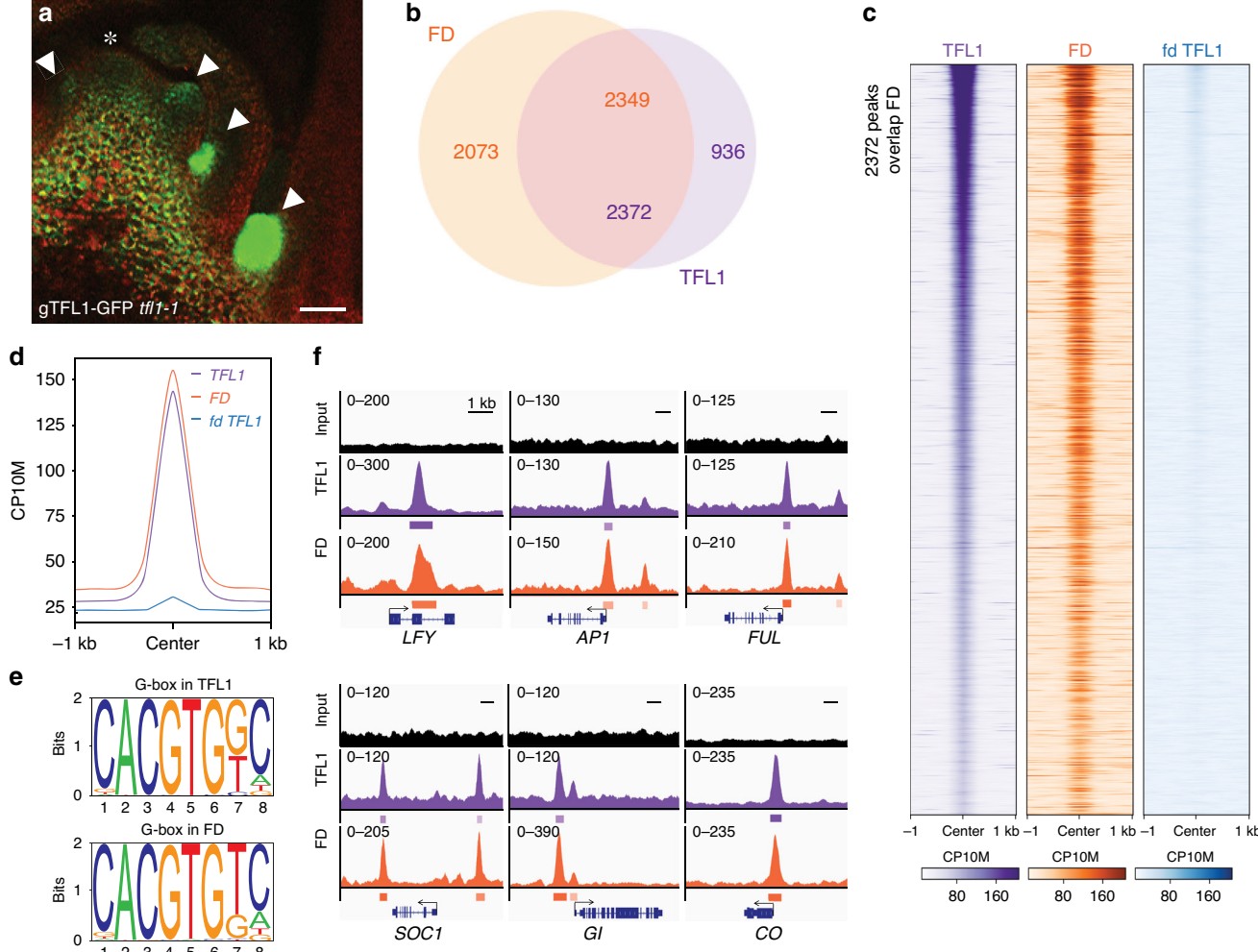

**Fig. 1 TFL1 is recruited by FD to target loci. a** gTFL1-GFP expression in branch meristems (arrowheads) in the axils of cauline leaves in 42-day-old short-day grown plants. Asterisk: shoot apex. **b** Overlap of significant (MACS2 $q$ value $\leq 10^{-10}$) FD and TFL1 ChIP-seq peaks. Numbers in the intersect show FD peaks overlapping with TFL1 peaks (orange) and TFL1 peaks overlapping with FD peaks (purple). **c, d** Comparison of ChIP-seq datasets. Heatmaps of TFL1, FD and TFL1(fd) ChIP-seq peaks (**c**). All heatmaps are centred on TFL1-binding peak summits and ranked from the lowest to the highest TFL1 summit signal. CP10M: counts per 10 million reads. ChIP-seq signal at TFL1 summits for TFL1 in the wild-type or in *fd* mutants and for FD (**d**). **e** Most significant motifs identified by de novo motif analysis under TFL1 or FD peak summits. **f** Browser view of TFL1 and FD-binding peaks at genes that promote onset of reproductive development[3] (*SOC1, GI, CO*) or the switch to flower fate[32] (*LFY, AP1, FUL*). Significant peaks (summit $q$ value $\leq 10^{-10}$) according to MACS2 are marked by horizontal bars, with the colour saturation proportional to the negative log 10 $q$ value (as for the narrowPeak file format in ENCODE). See also Supplementary Figs. 1–3 and Supplementary Data 1.

partially conserved C-boxes (Fig. 2b). When we transformed gTFL1-GFP *tfl1*-1 with a version of LFY e2 in which the three bZIP binding were mutated (e2m3), we were unable to detect TFL1 binding to the introduced copy of e2m3 (Fig. 2a). The three bZIP-binding sites in the second exon of *LFY* are thus necessary for TFL1 recruitment. Similar results were obtained when we tested TFL1 recruitment to e2 and e2m3 via FD in yeast (Supplementary Fig. 6c).

Having identified the *cis* motifs necessary for TFL1 recruitment to *LFY*, we next probed their contribution to spatiotemporal *LFY* accumulation. *LFY* reporters that contain e2 (pLFYi2-GUS, Supplementary Fig. 6a) and a genomic *LFY* reporter (gLFY-GUS, Fig. 2b) recapitulated endogenous LFY expression (Fig. 2c, Supplementary Fig. 6d)[34,35]. Mutating the three bZIP-binding sites in pLFYi2-GUS or gLFY-GUS caused ectopic reporter expression in the centre of the inflorescence shoot apex (Fig. 2c, Supplementary Fig. 6d). This is the precise region where TFL1 protein accumulates during reproductive development (Fig. 2c)[14]. Indeed, *LFY* is known to be ectopically expressed in the

inflorescence shoot apex of *tfl1* mutants during reproductive development[13]. Thus, TFL1–FD binding to the bZIP motifs of e2 of LFY is required to prevent ectopic *LFY* accumulation in the centre of the shoot apex.

Surprisingly, the bZIP-binding site mutations in the second exon of *LFY* in addition strongly reduced reporter expression in incipient and young flower primordia (Fig. 2c, Supplementary Fig. 6d). This suggests that the bZIP motifs may be required for *LFY* upregulation in these flower primordia, perhaps via FT. Based on prior studies[31,38–41], *LFY* was not thought to be an immediate early FT target. Because constitutive mutants that delay onset of the reproductive phase, like *ft*, indirectly delay the switch to flower formation[42] we wished to deplete FT specifically during the reproductive phase to test whether FT promotes *LFY* expression. Towards this end, we used a minimal *FT* promoter (p4kbFT) that is active in parts of leaves and stems in long-day grown plants only after day 12[43]. We fused p4kbFT to a previously characterized FT-specific artificial microRNA[44]. In the resulting conditional *ft* mutant (p4kbFT:amiRFT), onset of

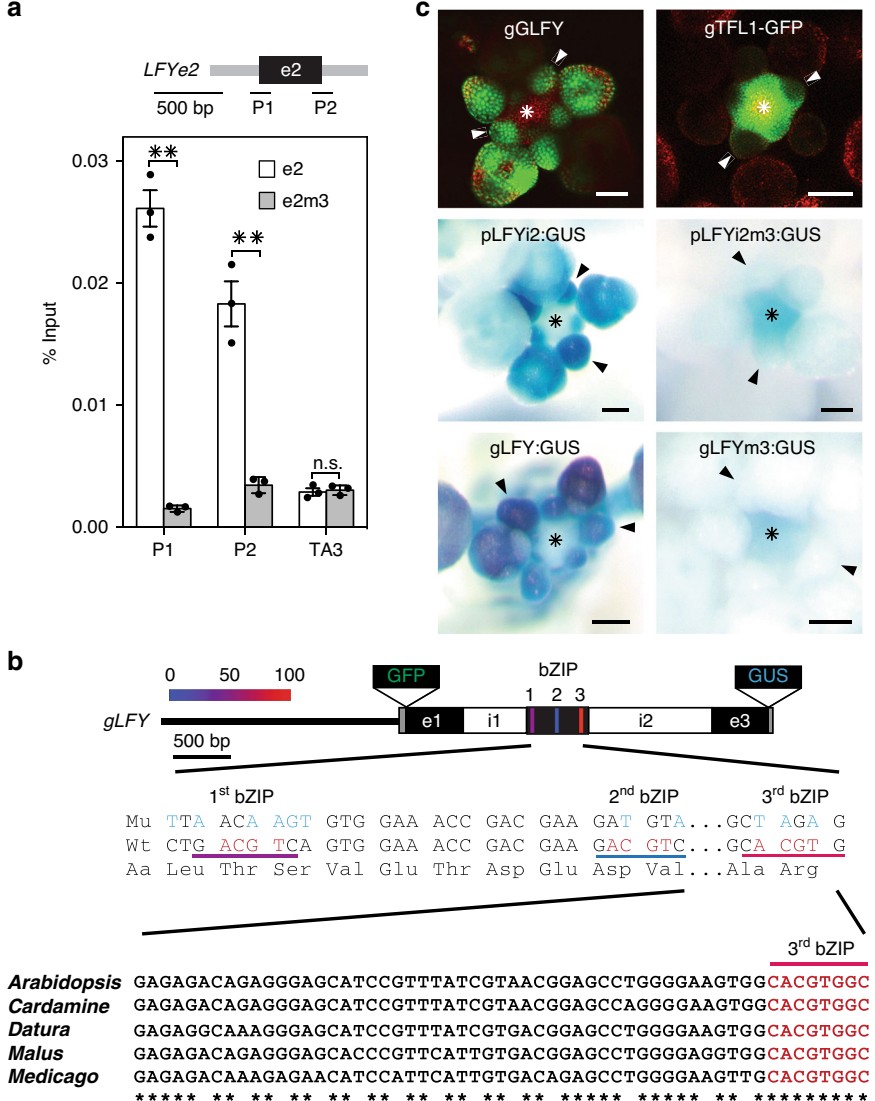

**Fig. 2 Exonic bZIP *cis* motifs mediate TFL1 recruitment to *LFY*. a** TFL1 recruitment to *LFY* exon 2 (e2) or a bZIP-binding site mutated version thereof (e2m3) in 42-day-old short-day-grown plants. Top: LFY exon 2 (e2, black rectangle) and T-DNA vector (grey line). Centre: location of amplicons P1 and P2, each consisting of one exon 2-specific and one vector-specific primer. Bottom: gTFL1-GFP ChIP-qPCR. Progeny pools of >50 random gTFL1-GFP T1 plants transformed with e2 or e2m3 were analyzed. Shown are mean ± SEM of three independent biological experiments (black dots). *P* values unpaired one-tailed *t*-test: **P1 = 0.0018, **P2 = 0.0078; n.s. TA3 = 0.35. **b** Top: genomic *LFY* construct with GFP (gGLFY) or beta-glucuronidase (gLFY-GUS). Putative bZIP-binding motifs in *LFY* exon 2 are colour-coded based on conservation. For the pLFYi2:GUS reporter diagram see Supplementary Fig. 6a. Centre: mutation of the three bZIP-binding motifs without changing the primary amino acid sequence. Bottom: evolutionary conservation of the G-box (3rd bZIP). **c** Top: expression domain of LFY and TFL1 proteins in inflorescence apices with flower primordia. Centre and Bottom: accumulation of beta-glucuronidase in a wild-type or bZIP-binding site mutated (m3) reporter (pLFYi2:GUS) or genomic construct (gLFY-GUS). Plants were grown in long day. Staining was conducted under identical conditions. Arrowheads: flower primordia; asterisk: inflorescence shoot apex; Scale bars: 2 mm. See also Supplementary Figs. 4–6.

reproduction was not delayed (Fig. 3a, Supplementary Fig. 7). However, p4kbFT:amiRFT plants displayed a significantly delay in onset of flower formation and failed to upregulate *LFY* expression (Fig. 3a and Supplementary Fig. 7).

In a parallel approach to test the role of *FT* in LFY induction, we induced endogenous *FT* expression by treating 42-day-old short-day grown plants with a single far-red-enriched long-day photoperiod (FRP). Far-red light enhances *FT* induction by photoperiod (Supplementary Fig. 8a, b)[45,46]. In addition, FRP triggered significant *LFY* induction, which was dependent on the presence of FT (Supplementary Fig. 8c, d). A single FRP treatment also induced the gLFY-GUS reporter, but only if the bZIP-binding sites in e2 were intact (Fig. 3b). Finally, we probed for rapid *LFY* induction by FT after generating an estradiol inducible version of

FT (35S:FT-HA^ER) (Supplementary Fig. 9a–c). A single steroid treatment triggered significant *LFY* induction after 4 h (Supplementary Fig. 9d). After crossing FT-HA^ER to LFY reporters containing (pLFYi2:GUS) or lacking (pLFYi2m3:GUS) the bZIP-binding sites in e2, we tested reporter activity in response to steroid activation. GUS upregulation was similar to that of endogenous *LFY* when the bZIP-binding sites were present (Supplementary Fig. 9e). By contrast, GUS expression was not upregulated in the pLFYi2m3:GUS FT-HA^ER plants after steroid induction (Supplementary Fig. 9e). The combined loss-of-function, photoinduction and gain-of-function data indicate that FT–FD directly activates *LFY* expression via bZIP-binding sites in the second exon.

These findings prompted us to assess the biological importance of the *LFY* bZIP-binding sites for inflorescence architecture.

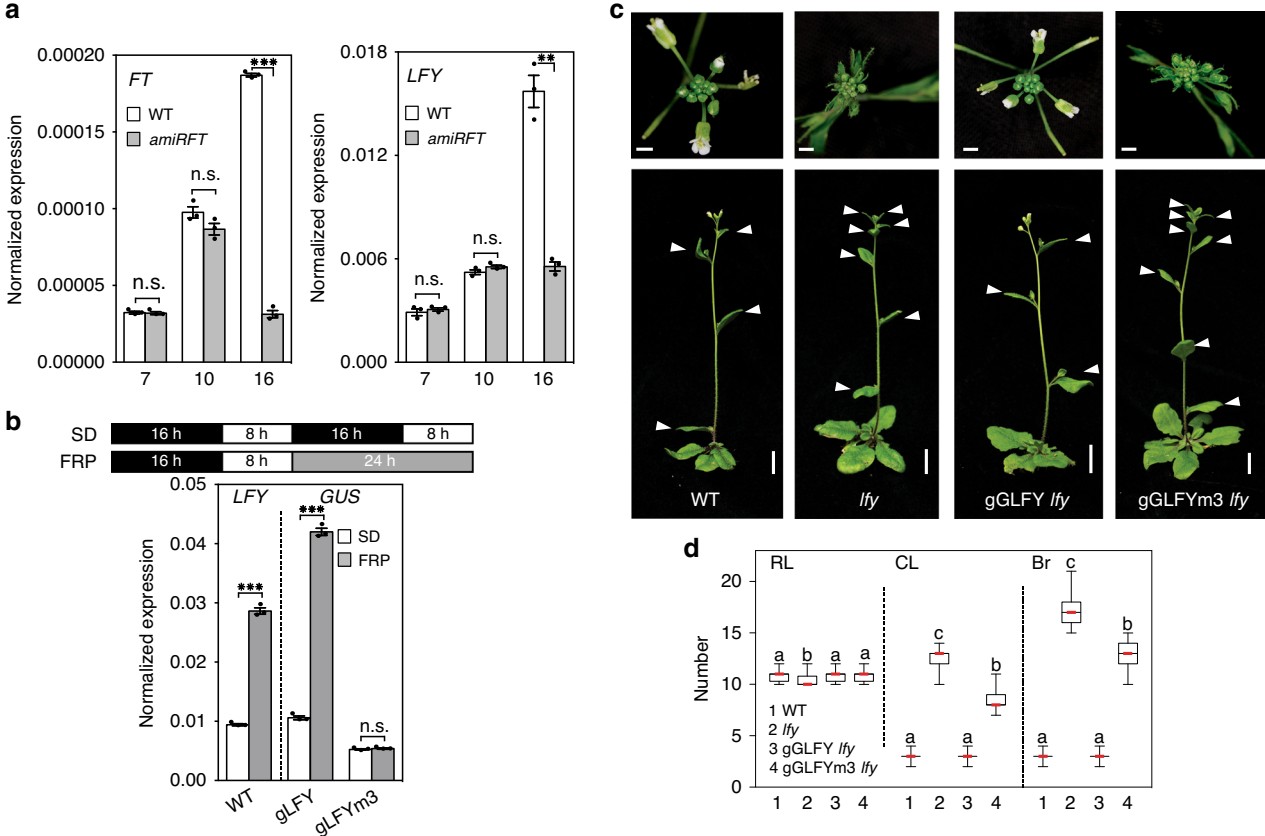

**Fig. 3 FT upregulates *LFY* expression to promote floral fate. a** Expression of *FT* and *LFY* in above-ground tissues of long-day-grown wild-type (WT) and *pFT4kb:amiRFT* plants prior to onset of (day 7 and day 10) or during (day 16) reproductive development. **b** Effect of a single far-red enriched photoperiod (FRP) on *LFY* (left) and reporter (*GUS*, right) accumulation in 42-day-old short day (SD) grown plants. **a**, **b** Expression was normalized over *UBQ10*. Shown are mean ± SEM of three independent biological experiments (black dots). Unpaired one-tailed *t*-test; *p* values: n.s. *FT* day 7 = 0.407, n.s. *FT* day 10 = 0.052, ***FT* day 16 = 6E−05; n.s. *LFY* day 7 = 0.258, n.s. *LFY* day 10 = 0.07, ***LFY* day 16 = 0.0045 (**a**) ****LFY* WT ± FRP = 0.0004, ***gLFY-GUS ± FRP = 0.0001, n.s. gLFY-GUSm3 ± FRP = 0.217 (**b**). **c** Rescue of *lfy-1* null mutants by genomic GFP-tagged LFY (gGLFY) or a bZIP-binding site mutated version thereof (gGLFYm3) in long-day-grown plants. Representative inflorescence images (top and side view). Arrowheads indicate branches formed on the main stem. For m3 mutations see Fig. 2a. **d** Phenotype quantification of 15 independent transgenic lines for gGLFY *lfy* and gGLFYm3 *lfy*. RL rosette leaves, CL cauline leaves, Br branches. Box plot-median (red line), upper and lower quartiles (box edges), and minima and maxima (whiskers). Letters: significantly different groups *p* value < 0.05 based on Kruskal–Wallis test with Dunn's *post hoc* test. Scale bars, 1 cm. See also Supplementary Figs. 6–10.

While a genomic GFP-tagged *LFY* construct (gGLFY) fully rescued the *lfy-1* null mutant (in 24 out of 25 independent transgenic lines), a construct which preserves LFY protein sequence but has mutated bZIP-binding sites (gGLFYm3) yielded only partial rescue (in 15 out of 15 independent transgenic lines) (Fig. 3c, d and Supplementary Fig. 10). In the gGLFYm3 *lfy-1* plants onset of flower formation was significantly delayed, leading to formation of many more branches, and *LFY* accumulation was strongly reduced (Fig. 3c, d and Supplementary Fig. 10). The dramatic reduction of *LFY* accumulation is striking given the many additional positive inputs into *LFY* upregulation previously identified[38,40,47,48]. Our combined data uncover a pivotal role of FT in *LFY* upregulation and reveal that FT promotes flower formation via *LFY*. We note that *LFY* accumulation in the centre of the gGLFYm3 *lfy-1* shoot apex was much lower than that observed in *tfl1* mutants and that gGLFYm3 *lfy-1* did not exhibit the terminal flower phenotype typical of *tfl1* (Supplementary Fig. 10)[13]. This is expected since the bZIP mutations at the *LFY* locus prevent access of both TFL1 and of activating PEBP family members FT and the closely related TWIN SISTER OF FT (TSF). Indeed, it has been shown that the terminal flower phenotype of *tfl1* is suppressed in *tfl1 ft tsf* triple mutants[49].

**FT competes TFL1 from FD bound at shared target loci.** Having identified *LFY* as a target under dual opposite regulation by TFL1 and FT, we next investigated the mechanism underlying the TFL1–FT antagonism at this locus. To test for possible competition between FT and TFL1 at the chromatin, we conducted anti-HA ChIP-qPCR in 42-day-old short-day grown FT-HA^ER gTFL1-GFP plants four hours after mock or steroid application. Estradiol induction led to rapid recruitment of FT-HA to the second exon of *LFY*, the region occupied by FD and TFL1 (compare Fig. 4a, b to Supplementary Fig. 4h). Anti-GFP ChIP-qPCR performed on the same sample uncovered a concomitant reduction in TFL1 occupancy (Fig. 4a, b). We next asked whether upregulation of endogenous FT also triggers reduced TFL1 occupancy at the *LFY* locus. Towards this end, we treated plants with a single FRP to upregulate FT (Supplementary Fig. 8b). The single FRP likewise significantly reduced TFL1 occupancy at the *LFY* chromatin (Fig. 4c). By contrast, photo-induction of FT did not alter FD occupancy (Fig. 4d). To probe whether FT is recruited to the second exon of LFY via the bZIP-binding sites, we transformed FT-HA^ER with either a wild-type version of e2 or a bZIP-binding site mutated version thereof (e2m3). After steroid induction, we used transgene-specific

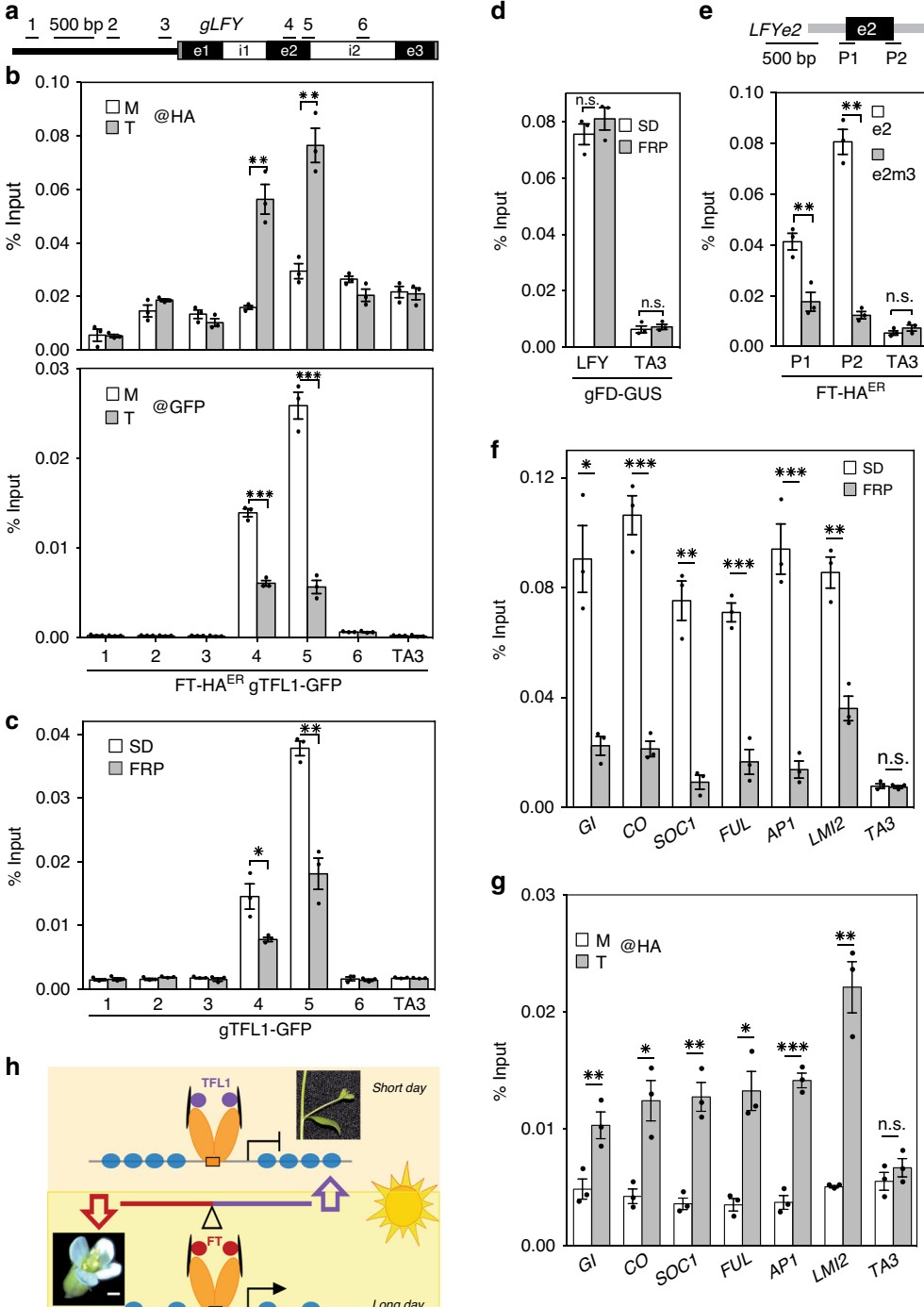

**Fig. 4 FT competes TFL1 from the chromatin. a** *LFY* locus and primers used. **b** Top: FT-HA[ER] occupancy at the *LFY* locus after 4-h mock (M) or steroid (T) treatment. Bottom: gTFL1 occupancy in the same sample. **c**, **d** Effect of *FT* upregulation by photoperiod (FRP, 24 h) on TFL1 (**c**) or FD (**d**) occupancy at the *LFY* locus. **e** FT-HA[ER] recruitment to *LFY* exon 2 (e2) or a bZIP-binding site mutated version thereof (e2m3) after 4-hour steroid treatment. Top: LFY exon 2 (e2, black rectangle) and T-DNA vector (grey line). Centre: location of amplicons P1 and P2, each consisting of one exon 2-specific and one vector-specific primer. Below: FT-HA[ER] ChIP-qPCR. Progeny pools of >50 random FT-HA[ER] T$_1$ plants transformed with e2 or e2m3 were analyzed (see also ref. [108]). **f** Effect of photoperiod (FRP) on TFL1 occupancy at TFL1–FD target loci identified in Fig. 1f and *LMI2*[67] **g** FT-HA[ER] occupancy with or without 4-h estradiol treatment at TFL1–FD bound regions of target loci shown in **f**. **b**–**g** ChIP was performed in 42-day-old short-day-grown plants. Shown are mean ± SEM of three independent biological experiments (black dots). *P* values (unpaired one-tailed *t*-test): **b** anti-HA ChIP ** region 4 = 0.009, region 5 = 0.003; anti-GFP ChIP ***region 4 = 6E−05, region 5 = 0.006; **c** *region 4 = 0.039, **; region 5 = 0.002; **d** n.s. LFY = 0.18, TA3 = 0.29. **e** **P1 = 0.004, P2 = 0.002, n.s. TA3 = 0.12; **f** * *GI* = 0.016, ****CO* = 0.0007, ***SOC1* = 0.0015, ****FUL* = 0.0003, ****AP1* = 0.007, *LMI2* = 0.0011, n.s. *TA3* = 0.34. **g** ***GI* = 0.009, **CO* = 0.011, ***SOC1* = 0.003, **FUL* = 0.015, ****AP1* = 0.0001, ***LMI2* = 0.008, n.s *TA3* = 0.17. **h** Model for antagonistic roles of TFL1 (purple circles) and FT (red circles) in promoting branch fate or floral fate, respectively. Increased FT accumulation leads to competition of TFL1 from bZIP transcription factor FD bound to chromatin and to onset of flower formation. FD dimers (orange ovals), 14-3-3 proteins (black disks).

primers to monitor FT binding to the two versions of *LFY* e2 by ChIP-qPCR. As described above for TFL1 (Fig. 2a), FT was recruited to *LFY* e2 alone (Fig. 4e). In addition, FT recruitment to the introduced copy of *LFY* e2 was abolished when the three bZIP-binding sites were mutated (Fig. 4e).

Our combined data suggest that FT competes TFL1 from FD bound at exonic bZIP motifs at the *LFY* locus. The loss of TFL1 from the LFY locus via competition by FT is further supported by the finding that neither steroid nor FRP induction of FT reduced *TFL1* mRNA accumulation (Supplementary Figs. 8c and 9d).

Competition of TFL1 from FD by FT is not limited to the *LFY* locus. We tested whether FT induction by FRP competes TFL1 from the other direct TFL1–FD target loci we identified (Fig. 1f). FRP treatment reduced TFL1 occupancy at all loci tested (Fig. 4f). To confirm that FT indeed occupies the TFL1–FD bound sites at these loci, we also conducted ChIP-qPCR in FT-HA[ER] after steroid induction. In the estradiol treated samples, we saw significant FT recruitment to the TFL1–FD bound regions at all loci tested (Fig. 4g). We conclude that the antagonism between FT and TFL1[19,21,27] relies on competition for FD bound at the chromatin of shared target loci (Fig. 4h).

**Direct TFL1–FD repressed genes promote onset of flower formation and endogenous signalling.** Our findings place florigens directly upstream of LFY, yet prior genetic data suggest that florigens act both upstream of and in parallel with LFY[39,50]. To gain insight into additional gene expression programs repressed by the TFL1–FD complex, we next conducted RNA-seq with and without FRP treatment. We isolated inflorescences with associated primordia from 42-day-old short-day-grown *ft* mutant, wild-type and *tfl1* mutant plants and identified the significant gene expression changes in each genotype relative to untreated siblings. On the basis of Principle Component Analysis (PCA) and replicate analysis, RNA-seq quality was high (Supplementary Fig. 11). We next defined genes directly repressed by TFL1–FD. Towards this end, we focussed on TFL1–FD complex bound loci that exhibit FT-dependent de-repression upon photoinduction. Six-hundred four TFL1–FD bound genes were significantly (DESeq2 adjusted $p < 0.005$) de-repressed upon FRP treatment in the wild-type or in *tfl1* mutants but not in *ft* mutants (Fig. 5a). GO term enrichment linked the TFL1–FD repressed genes to reproductive development and to response to endogenous and abiotic signals (Fig. 5b).

K-means clustering of the 604 genes identified three main patterns of gene expression. Genes encoding promoters of floral fate[32] (*LFY*, *AP1*, *FUL* and *LMI2*) clustered together (cluster III in Fig. 5c) and displayed stronger upregulation in *tfl1* mutants than in the wild type. This pattern of de-repression was confirmed for all four loci using independent biological samples and qRT-PCR (Supplementary Fig. 12a). *SOC1* clusters with these genes, but was not included in further analyses because it was weakly, but significantly, de-repressed in *ft* mutants (Fig. 5c, Supplementary Data 1 and Supplementary Fig. 12b). By contrast, *CO* and *GI*, which promote cessation of vegetative development[2,3], were more strongly upregulated in the wild-type than in *tfl1* mutants (cluster I in Fig. 5c), perhaps because these genes are already partially de-repressed in *tfl1* mutants in the absence of FRP treatment. Indeed, like RNA-seq, qRT-PCR of independent biological replicates revealed higher accumulation of *GI* and *CO* in untreated *tfl1* mutant compared to wild-type plants (Fig. 5c, Supplementary Data 1 and Supplementary Fig. 12b). Cluster II genes are only upregulated in the wild type and may represent genes that are transiently de-repressed. Our data identify the TFL-FD complex as a hub for repression of key regulators of the onset of reproductive development and of the switch to flower fate (Fig. 5c).

Consistent with the GO-term enrichment analysis (Fig. 5b), combined ChIP-seq and RNA-seq analysis additionally identified components of endogenous stimulus response. We identified genes linked to sugar signalling (trehalose-6-phosphate) and hormonal signalling and response (abscisic acid, cytokinin, brassinosteroid, auxin and strigolactone) as direct TFL1–FD complex repressed targets (Fig. 5c). Using qRT-PCR and independent biological samples, we confirmed FT-dependent de-repression of members of these pathways by FRP photoinduction (Fig. 5c). Several of the identified pathways link to repression of branching or to promotion of onset of flower formation in the inflorescence. For example, we identified four trehalose-6-phosphate phosphatases (*TPPH*, *TPPJ*, *TPPG* and *TPPE*) as direct TFL1–FD complex repressed targets; TPPs were recently shown to repress branching in the maize inflorescence[51]. In addition, auxin and the auxin-activated transcription factor *MONOPTEROS* (*MP*) were direct TFL1–FD complex repressed targets (Fig. 5c). MP promotes the switch to floral fate in *Arabidopsis*[52] and the tomato ortholog of TFL1 executes its role in inflorescence architecture at least in part by modulating auxin flux and response[53]. Finally, we identified key components of the brassinosteroid pathway including the *BRI1* receptor and the bHLH transcription factor *BIM1*[54,55] as direct TFL1–FD complex repressed targets. Brassinosteroid signalling represses inflorescence branching in Setaria[56]. The combined data implicate TFL1–FD in direct repression of genes that promote floral fate or repress branch fate, consistent with the role of TFL1 in promoting branch formation.

We also identified the cytokinin activating enzyme *LOG5*[57], abscisic acid biosynthesis (*ABA1*) and response regulators (*ABI5*, *ABF4*, *APF2*)[58,59], and components of strigolactone signalling, *SMXL6* and *SMXL8*[60,61], as direct TFL1–FD complex repressed targets.

To gain further insight into the role of the TFL1–FD complex in hormone signalling, we next assessed indirect, downstream, gene expression changes triggered by FRP treatment. In particular, we identified genes not bound by TFL1 or FD that were significantly differentially expressed (DESeq2 adjusted $p < 0.005$) in the wild type and in *tfl1*, but not in *ft*. The identified indirect targets provide a 'molecular phenotype' that is consistent with de-repression of the auxin, brassinosteroid and cytokinin hormone pathways upon FRP treatment in the wild-type and in *tfl1* mutants (Fig. 5c). By contrast, the abscisic acid signalling pathway signature was more complex (Fig. 5c). Our combined data uncover a prominent role for the TFL1–FD complex in regulation of endogenous signalling.

It is conceivable that components of some of the identified TFL1–FD dependant pathways (strigolactone, cytokinin, auxin, abscisic acid as well as sugar signalling) may modulate additional aspects of the inflorescence architecture, such as branch out-growth[62–64]. Support for this hypothesis comes from our phenotypic analyses. We examined the effect of a single FRP on inflorescence architecture in the *ft* mutant, the wild-type and the *tfl1* mutant. Photoperiod induction triggered a reduction in the number of branches, but not cauline leaves, formed in the wild type and more strongly, in *tfl1* (Supplementary Fig. 13a–j). This suggests that branch meristems adopt floral fate upon stimulus perception[65]. *tfl1* mutants also formed fewer branches than the wild type in the absence of photoperiod. These phenotypes are consistent with the observed gene expression changes (Fig. 5c). In addition, FRP triggered a significant increase in inflorescence branch outgrowth in both wild-type and *tfl1* plants (Supplementary Fig. 13k). FRP had no phenotypic effect in *ft* mutants. Our combined data suggest that florigens tune plant form to the environment by controlling expression of master developmental regulators and endogenous signalling pathway components. These developmental changes likely require large-scale transcriptional reprograming in the context of chromatin. Consistently, we identified transcriptional co-

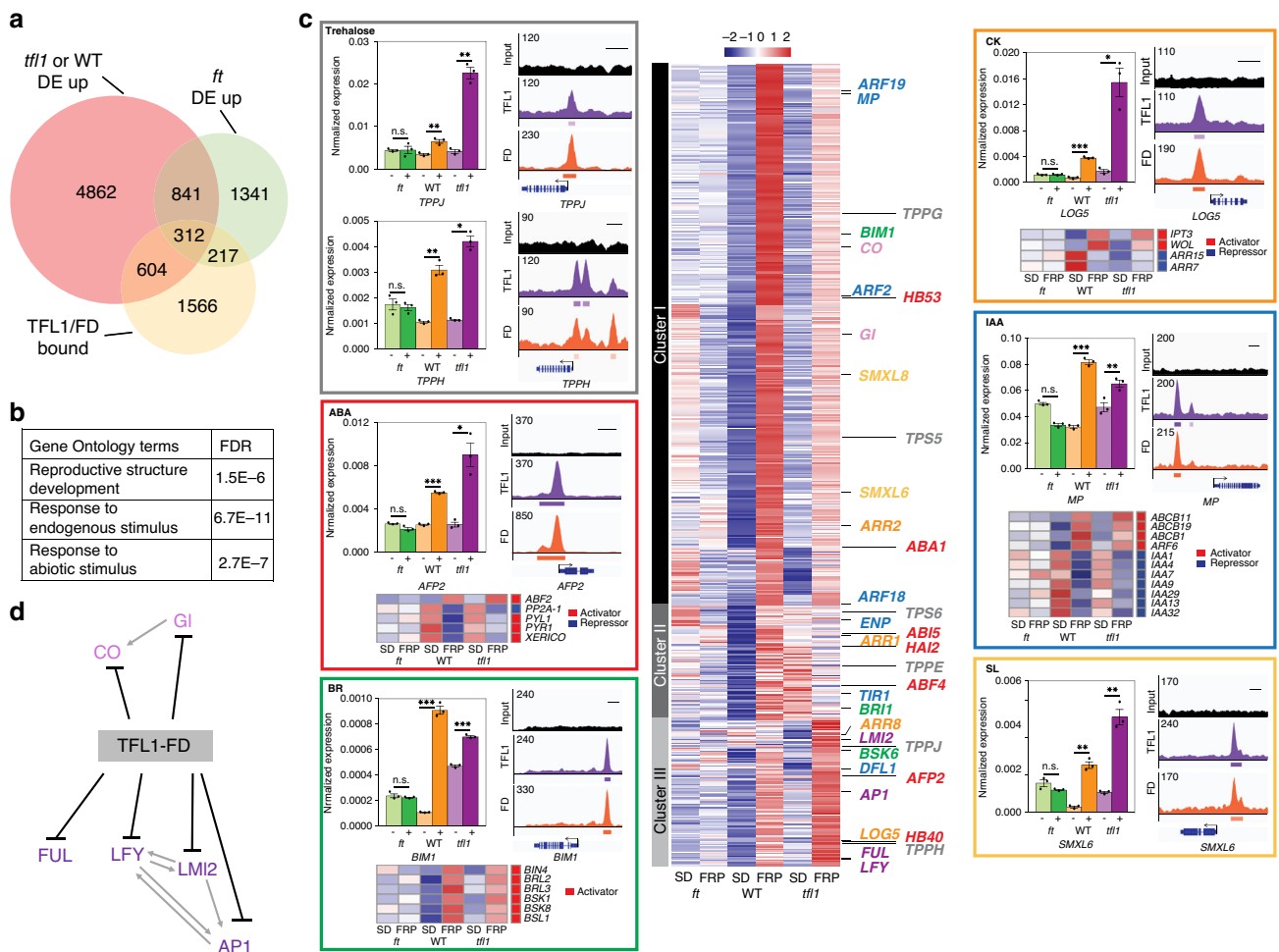

**Fig. 5 Genes directly repressed by the TFL1–FD complex. a** Six-hundred four genes bound by TFL1 and FD are de-repressed after a single FRP specifically in the wild-type or in *tfl1* mutants. **b** Gene ontology term enrichment (Goslim AgriGO v2) for the 604 direct TFL1–FD complex repressed genes. **c** K-means clustering of the direct TFL1–FD complex repressed genes (centre). Heatmap with gene names colour coded for the pathway they act in. Onset of reproduction (light purple), floral fate (purple), sugar signalling (grey), abscisic acid (ABA, red), brassinosteroid (BR, green), cytokinin (CK, orange), auxin (IAA, blue) and strigolactone (SL, yellow). In colour-coded boxes: Left: Independent confirmation of gene expression changes of direct TFL1–FD targets by qRT-PCR of plants with (+) or without (−) a single FRP. Shown are mean ± SEM of three independent biological replicates (black dots). *P* = values (unpaired one-tailed t-test) *TPPJ*: n.s. *ft* = 0.45, **WT = 0.0044, **tfl1* = 0.0027; *TPPH*: n.s. *ft* = 0.99, ** WT = 0.004, **tfl1* = 0.002; *AFP2*: n.s. *ft* = 0.96, ***WT = 2E−06, **tfl1* = 0.01; *BIM1*: n.s. *ft* = 0.76, ***WT = 0.0007, ***tfl1* = 1.7E−04; *LOG5*: n.s. *ft* = 0.44, ***WT = 8.7E−06, *tfl1* = 0.012; *MP*: n.s. *ft* = 0.99, ***WT = 7.2E−05, **tfl1* = 0.007; *SMXL6*: n.s. *ft* = 0.91, **WT *p* value = 0.003, **tfl1* = 0.005. Right: ChIP-seq-binding screenshots. Below: (where applicable) heatmaps of indirect (downstream) FT-dependent gene expression changes of known pathway activators (red in sidebar) or repressors (blue in sidebar). Indirect target genes include: abscisic acid: upregulation of some positive regulators of abscisic acid response (*ABF2*) and repression of others (*PYL1, PYR1, XERICO*)[58,59]; Brassinosteroid: upregulation of multiple positive signalling pathway components (*BRL2, BRL3*, BSK1, *BSK8, BSL1* and *BIN4*)[123]. Cytokinin: upregulation of cytokinin production (*IPT3*) and perception (*WOL*) and downregulation of negative cytokinin response regulators (*ARR7, ARR15*)[124]. Auxin: upregulation a positive auxin response regulator (*ARF6*) and downregulation of negative auxin response regulators (*IAAs*)[125]. **d** Interaction network of TFL1–FD complex repressed targets linked to onset of reproduction (CO, GI) and floral fate (all others). Previously described interactions[2,3,32] are indicated by grey arrows. See also Supplementary Figs. 11–14.

regulators and chromatin regulators among the direct TFL1–FD repressed targets (Supplementary Fig. 14a, b).

## Discussion

Here we identify *LFY*, a master regulator of flower fate[33,34], as a target under dual opposite transcriptional regulation by TFL1 and FT and demonstrate that FT activation of *LFY* expression is critical to promote floral fate. We provide a molecular framework for the antagonistic roles[5,27] of FT and TFL1 that relies on competition for bZIP transcription factor mediated access to binding sites at regulatory regions of shared target loci. Additional support for this mechanism comes from recent in vitro studies[21].

Our data suggest that TFL1 may not simply prevent access of the FT co-activator to the chromatin[23] but may be an active repressor, as mutating bZIP-binding sites results in *LFY* de-repression specifically in the TFL1 expression domain. The identity of the transcription factors that activate *LFY* in the centre of the inflorescence shoot apex in the absence of PEBP/FD binding to *LFY* is not known. Our identification of FT recruiting motifs in the second exon of *LFY* fits with prior data demonstrating that the 2.3 kb upstream intergenic '*LFY* promoter' is unresponsive to FT[38]. This upstream regulatory region drives reporter expression in similar domains as endogenous *LFY*[66]. The requirement of the bZIP motifs for *LFY* upregulation in the context of the genomic construct, which contains the 2.3 kb '*LFY*

promoter', suggests the presence of repressive regulatory elements in the genic region of *LFY*.

We identify hundreds of TFL1–FD repressed genes many of which, based on our computational analyses of recently published FD and TFL1 ChIP-seq datasets[17,30], are also immediate early gTFL1 or gFD targets in long-day conditions. Eighty two percent of our 604 high-confidence TFL1–FD repressed genes are present in at least one of the long-day ChIP-seq datasets (Supplementary Fig. 3b). The 604 direct TFL1–FD repressed genes include key regulators of onset of the reproductive phase and of floral fate. Of note, TFL1 opposes not only LFY, but also LFY targets, such as *LMI2* and *AP1*[67,68]. This is consistent with prior genetic investigations that place TFL1 both upstream of LFY and as a modulator of plant response to LFY[50]. Finally, we link the TFL1–FD complex to repression of diverse endogenous signalling pathways including sugar and hormonal signals. Several of these pathways have been shown to impact the switch from branch or flower fate in other plant species[51,53,56]. The combined data point to an important role of the hormonal environment for the switch from branch to flower fate in primordia of the inflorescence. Our findings also set the stage for elucidating communalities as well as differences between florigen regulated cell fate reprogramming during flower initiation and other developmental pathways under seasonal control by florigens such as tuberization, bulb formation and seed dormancy[69–72].

Changes in the relative balance of activating and repressive PEBP family members occurred during domestication of diverse crop species to give rise to desirable traits like everbearing and compact growth habits[5,8,69,73]. Thus, mechanistic insight into the antagonism and identification of the targets of PEBPs will benefit traditional or genome editing-based crop improvement. It should further facilitate elucidation that how PEBP protein act as co-activators or co-repressors in the nucleus.

## Methods

**Plant materials.** *Arabidopsis* ecotype Columbia plants were grown in soil at 22 °C in long-day photoperiod (LD, 16 h light/8 h dark, 100 μmol/m² s) or short-day photoperiod (SD, 8 h light/16 h dark, 120 μmol/m² s). gFD-GUS[20] and fd-1 null mutants[74], *tfl1-14* hypomorph mutant[27,75], *tfl1-1* severe mutant[27,75,76], *lfy-1* null mutant[33,77,78], *ft-10* null mutant[79], 35S:LFY[68] and 35S:TFL1 (ref. [35]) were previously described. 35S:LFY (Landsberg *erecta*) was introgressed into the Columbia background through backcrossing. gFD-GUS[80] was crossed into the *fd-1* null mutant background.

**Constructs for transgenic plants.** For *gTFL1-GFP*, GFP followed by a peptide linker (GGGLQ) was fused to an 8.4 kb BamHI (NEB, R0136S) genomic fragment from lambda TFG4 (ref. [81]). This fragment was introduced into the binary vector *pCGN1547* (ref. [82]). For TFL1ER and FT-HAER, TFL1 and FT were PCR amplified from cDNA; in the case of FT, the 3′ primer contained three times Hemagglutinin (HA) plus a stop codon. PCR products were cloned into *pENTRD-TOPO* (Invitrogen, K243520) and shuffled into *pMDC7* (ref. [83]) by LR reaction (Invitrogen, 11791-020).

For *LFY-GUS* reporters, the bacterial beta-glucuronidase (GUS) gene from the *pGWB3* (ref. [84]) binary vector was fused to *pENTRD-TOPO* vector containing the 2290-bp *LFY* promoter[66] alone (pLFY:GUS), the *LFY* promoter and *LFY* genic region up to and including the first intron (pLFYi1:GUS), or the *LFY* promoter and *LFY* genic region up to and including the second *LFY* intron (pLFYi2:GUS) by LR reaction. bZIP-binding site mutations in *LFY* e2 (pLFYi2m3: GUS) were generated by Ω-PCR[85]. gGLFY was constructed by PCR amplifying a 4929-bp genomic *LFY* fragment (gLFY), including the 2290-bp *LFY* promoter, from genomic DNA followed by cloning into the KpnI-HF (NEB, R3124S) and NotI-HF (NEB, R3189S) digested *pENTR3C* vector by Gibson Assembly. Next, GFP was inserted at position + 94 bp, as previously described for *pLFY:GLFY*[42,80], by Ω-PCR[85]. bZIP-binding site mutations were introduced into *pENTR3C-gGLFY* by Ω-PCR to generate gGLFYm3. Both constructs were shuffled into *pMCS:GW*[86] using LR reaction. To create gLFY:GUS, the 4929-bp genomic *LFY* clone minus the stop codon and the GUS fragment were PCR amplified and inserted into linearized *pENTR3C* by Gibson Assembly. For gLFYm3:GUS, bZIP-binding site mutations were introduced into *pENTR3C-gLFY:GUS* by Ω-PCR. To test recruitment of TFL1 and FT to e2 of *LFY*, wild-type (e2) or bZIP-binding site mutated exon 2 (e2m3) were PCR amplified and cloned into *pGWB3* (ref. [84]).

For test of recruitment, LFY e2 and e2m3 were amplified by forward (5′-caccAA CAGCAGCAGAGACGGAGAAAGAA-3′) and reverse (5′-TCGTACAAGTGGA ACAGATAATC-3′) primers and cloned into pGWB3 binary vectors, which were transformed into gTFL1-GFP and 35S:FT-HAER.

For *pFT4kb:amiRFT*, a 3994-bp truncated *FT* promoter[87] was PCR amplified from genomic DNA as was the published *amiRFT*[44] from *pRS300* (ref. [88]). The *amiRFT* fragment was introduced into EcoRI-HF (NEB, R3101S) digested *pENTR3C* (Thermo Fisher Scientific, A10464) by Gibson Assembly (NEB, E5510S) and shuffled into binary vector *pMCS:GW*[86] using LR reaction, which resulted in pMCS:amiRFT. The previously described 3994-bp *FT* promoter was inserted to XhoI (NEB, R0146S) digested pMCS:amiRFT by Gibson Assembly.

Genomic DNA was extracted using the GenElute Plant Genomic DNA Miniprep Kit (Sigma-Aldrich, G2N70). Primer sequences are listed in Supplementary Table 2. All constructs were sequence verified prior to transformation into plants with Agrobacterium strain GV3101 by floral dip[89]. Plant lines generated are listed in Supplementary Table 3.

**Imaging.** Images were taken with a Canon EOS Rebel T5 camera for plant phenotypes and yeast one-hybrid assays, or with a stereo microscope (Olympus SZX12) equipped with a colour camera (Olympus LC30) for GUS images and inflorescence phenotypes. For GFP images, a Leica TCS SP8 Multiphoton Confocal with a 20× objective was used with a 488 nm excitation laser and emission spectrum between 520 and 550 nm (GFP) or 650–700 nm (chlorophyll autofluorescence) using standard imaging techniques[90,91]. For *tfl1-1* gTFL1-GFP in short-day photoperiod, shoot apices were sectioned longitudinally on an oscillating tissue slicer (Electron Microscopy Sciences, OTS-4000) after embedding in 5% Agar (Fisher Scientific, DF0812-07-1).

**Plant treatment and gene expression analysis.** For test of gene expression, 16-day-old TFL1ER or 12-day-old FT-HAER plants grown in LD were induced by a single spray application of 10 μmol beta-estradiol (Sigma-Aldrich, 8875-250MG) dissolved in DMSO (Fisher Scientific, BP231-1L) and 0.015% Silwet L-77 (Plant-Media, 30630216-3). Mock solution consisted of 0.1% DMSO and 0.015% Silwet. To probe FT recruitment to and TFL1 occupancy at the *LFY* locus, 42-day-old FT-HAER gTFL1-GFP plants grown in SD were treated by a single spray application of 10 μmol beta-estradiol or mock solution. In all cases, tissues were harvested 4 h after treatment. To test for gain-of-function phenotypes in long-day conditions, FT-HAER, TFL1ER and FT-HAER gTFL1-GFP plants were treated with 10 μmol beta-estradiol or mock solution from 5-day onwards every other day until bolting.

FRP was applied at the end of the short day (ZT8) for 24 h using a Percival Scientific E30LED[45] with red (660 nm) to far-red (730 nm) ratio = 0.5 and light intensity 80 μmol/m² s. Control plants were kept in regular short-day conditions (16 h dark and 8 h light, red to far-red ratio = 12 and 120 μmol/m² s light intensity) for 24 h. Light intensity and spectral composition were measured by an Analytical Spectral Devices FieldSpec Pro spectrophotometer.

For qRT-PCR analysis, total RNA was extracted from leaves or shoot apices using TRIzol (Thermo Fisher Scientific) and purified with the RNeasy Mini Kit (Qiagen, 74104). cDNA was synthesized using SuperScript III First-Strand Synthesis (Invitrogen, 18080051) from 1 μg of RNA. Real time PCR was conducted using a cDNA standard curve. Normalized expression levels were calculated using the 2^(−delta delta CT) method with the housekeeping gene *UBQ10* (AT4G05320) as the control. Where expression of multiple different genes was compared, normalized gene expression is shown relative to the control treatment. Primer sequences are listed in Supplementary Table 2.

**Yeast one-hybrid assay.** LFY e2 and the bZIP-binding site mutated version (e2m3) were cloned into the KpnI-HF and XhoI linearized pAbAi vector (Takara) by Gibson Assembly and integrated into the yeast genome following the Matchmaker Gold Yeast One-Hybrid protocol (Takara) and the Y1Golden strain (Takara). Coding sequences of FD and TFL1 were cloned into the pENTRD-TOPO vector. After sequencing, constructs were shuffled into either pDEST32 or pDEST22 (Takara) by LR reaction and transformed into the DNA-binding region containing yeast strain. Empty pDEST32 and pDEST22 served as negative controls. Growth was assayed after serial dilution on growth media with or without 60 ng/ml Aureobasidin A (Clontech, 630499). Primer sequences are listed in Supplementary Table 2.

**ChIP-qPCR, ChIP-seq and data analysis.** Forty two-day-old short-day grown plants were trimmed and 1.6 g of non-bolted inflorescences were harvested from 36 plants. Chromatin immunoprecipitation was conducted following a published protocol[92] for ChIP-qPCR. For ChIP-seq, each biological replicate consisted of eight individual IP reactions pooled into one MinElute PCR (Qiagen, 28004) purification column. For ChIP-seq and ChIP-qPCR, anti-GFP antibody (Thermo Fisher Scientific, A-11122; 1:200 dilution) and anti-GUS antibody (Abcam, ab50148; 1:200 dilution) were used. The antibodies were validated by the manufacturers. ChIP-qPCR was performed using Platinum Taq DNA Polymerase (Invitrogen, 10966034) and EvaGreen dye (Biotium, 31000). For ChIP-qPCR, the value of the ChIP samples was normalized over that of input DNA as previously described[92]. Non-transgenic wild-type plants were used as the negative genetic

control for anti-GFP and anti-GUS antibody ChIP. The *TA3* retrotransposon (AT1G37110) was used as the negative control region for ChIP-qPCR. Primer sequences are listed in Supplementary Table 2.

Anti-GFP ChIP-seq was performed for gTFL1-GFP (A), gTFL1-GFP (B), *fd-1* gTFL1-GFP and control samples (non-epitope containing plants). Anti-GUS ChIP-seq was performed for *gFD-GUS*. Two biological replicates were sequenced in each case. Dual index libraries were prepared for the ten ChIP samples listed above and for four input samples using the SMARTer ThruPLEX DNA-Seq Kit (Takara Bio, R400406). Library quantification was performed with the NEBNext Library Quant Kit for Illumina (NEB, E7630). Single-end sequencing was conducted using High Output Kit v2.5 (Illumina, TG-160-2005) on the NextSeq 500 platform (Illumina).

FastQC v0.11.5 was performed on both the raw and trimmed[93] reads using TRIMMOMATIC v0.36[93] (ILLUMINACLIP:adaptors.fasta:2:30:10 LEADING:3 TRAILING:3 MINLEN:50) to confirm sequencing quality[94]. Reads with MAPQ ≥ 30 (SAMtools v1.7)[95] uniquely mapping to the Arabidopsis Information Resource version 10 (TAIR 10)[96] that were not flagged as PCR or optical duplicates by Bowtie2 v2.3.1(ref. [97,98]) were analyzed further by principal component analysis (PCA) using the plotPCA function of deepTools[99]. Reads were further processed following ENCODE guidelines[100], followed by cross-correlation analysis with the predict function of MACS2 (ref. [101]) to empirically determine the fragment length. Significant ChIP peaks and summits (summit *q* value ≤ $10^{-10}$) were identified in MACS2 for the pooled ChIP relative to the pooled negative controls (ChIP in non-transgenic wild type). Peak overlap (≥1 bp) was computed using BEDTools intersect v2.26.0 (ref. [102]) and statistical significance was computed using the hypergeometric test assuming a 'universe' of 10,000 possible peaks[103]. Heatmaps were generated using deepTools v3.1.2 (ref. [99]). The 3308 TFL1 and 4422 FD peaks were mapped to 3699 and 4493 Araport11 (ref. [104]) annotated genes, respectively, if the peak summit was intragenic or located ≤4 kb upstream of the transcription start site. Recently published datasets were analyzed in identical fashion. Genomic distribution of peak summits was called using the ChIPpeakAnno library[104,105].

De novo motif analysis was conducted using MEME-ChIP v5.0.2 (Discriminative Mode)[106] and HOMER v4.10 (ref. [107]) for MACS2 *q* value ≤ $10^{-10}$ peak summits (±250 base pairs) compared to genome-matched background (unbound regions from similar genomic locations as the peak summits) as previously described[108,109].

GO term enrichment analyses were performed using GOSlim in agriGO v2.0 (ref. [110]) and significantly enriched GO terms with *q* value < 0.0001 (FDR, Benjamini and Yekutieli method[111]) were identified.

**Correlation analyses**. Public ChIP-seq datasets for FD[30], for TFL1[17], LFY[112] and an unpublished LFY ChIP-seq dataset from our lab (GEO accession GSE141706) were analyzed as described above for TFL1 and FD ChIP-seq. To compare the relationship between all ChIP-seq datasets we calculated Pearson correlation coefficients for reads in regions of interest using deeptools v3.1.2[99]. Regions of interest were comprised of the combined significant peak regions (MACS2 ≤ *q* value $10^{-10}$) of all ChIP-seq datasets and read signal was derived from the sequencing-depth normalized bigwig file for each sample.

**RNA-seq and data analysis**. A single 24 h FRP was applied to 42-day-old short-day grown *ft-10* mutants, wild-type and *tfl1-1* mutants, starting at the end of the day (ZT8). After the treatment, 0.1 g of inflorescence shoots were harvested for each biological replicate after removing all leaves and roots. Three biological replicates were prepared for each experiment. RNA quantity and quality were analyzed by Qubit BR assay (Thermo Fisher Scientific, Q10210) and Agilent RNA 6000 Nano Kit (Agilent, 5067-1511) on an Agilent 2100 bioanalyzer, respectively. Libraries were constructed from 1 μg total RNA using the TruSeq RNA Sample Prep Kit (Illumina, RS-122-2001). After library quantification with the NEBNext Library Quant Kit for Illumina (NEB, E7630), single-end sequencing was conducted using the NextSeq 500 platform (Illumina).

RNA-seq analysis was conducted using FastQC v0.11.5 (ref. [94]) on raw sequences before and after trimming Trimmomatic v0.36 (ref. [93]) (ILLUMINACLIP:adapters.fasta:2:30:10 LEADING:3 TRAILING:3 MINLEN:50) to confirm sequencing quality[94]. Reads were mapped using the STAR mapping algorithm[113] (–sjdbOverhang 100 --outSAMprimaryFlag AllBestScore --outSJfilterCountTotalMin 10 5 5 5 --outSAMstrandField intronMotif --outFilterIntronMotifs RemoveNoncanonical --alignIntronMin 60 --alignIntronMax 6000 --outFilterMismatchNmax 2), to the TAIR 10 Arabidopsis genome-assembly[96], and Araport11 Arabidopsis genome-annotation[104]. Specific read coverage was assessed with HT-Seq[114] (--stranded = 'no' -minaqual = 30).

For PCA, raw read counts were subjected to variance stabilizing transformation and projected into two principal components with the highest variance[115–117]. In parallel, raw reads were adjusted for library size by DESeq2 v1.24.0 (ref. [118]). After PCA, two biological replicates per genotype and treatment were selected for further analysis. Gene normalized z-scores were used for k-means (MacQueen) clustering[119]. Pairwise differential expression analyses were performed by comparing FRP and untreated pooled normalized read counts in each genotype using default DESeq2 parameters with no shrinkage (ref. [118]) and an adjusted *p* value cut-off ≤ 0.005.

**Photoperiod shift phenotype analysis**. A single 24-h far-red light enriched photoperiod shift (FRP) was applied to 42-day-old short-day grown *ft-10* mutants, wild-type and *tfl1-1* mutants as for RNA-seq, followed by further growth in short-day conditions. To asses onset of reproductive development, the number of rosette leaves formed were counted at bolting. To analyze the inflorescence architecture, the number of sessile buds, outgrowing branches, flower branches, and single flowers subtended by a cauline leaf were counted weekly after bolting until the first normal flower (not subtended by a cauline leaf) formed.

**Statistical analyses**. The Kolmogorov–Smirnov (K–S) test[120] was used to assess whether the data were normally distributed. All ChIP and qRT-PCR data were normally distributed. An unpaired one-tailed *t*-test was used to test for changes in one direction. Error bars represent the standard error of the mean (SEM). Two to three independent biological replicates were analyzed. For multiple-group comparisons (phenotypes) the non-parametric Kruskal–Wallis test[121] followed by the Dunn's *post hoc* test[122] were employed. Box and whisker plots display minima and maxima (whiskers), lower and upper quartile (box) and median (red vertical line).

**Reporting summary**. Further information on research design is available in the Nature Research Reporting Summary linked to this article.

## Data availability
The authors declare that the data supporting the findings of this study are available within the paper, its Supplementary information files and public data repositories. Source data are provided with this paper. The ChIP-seq and RNA-seq datasets were deposited to the GEO database (GSE141894). Individual replicates and *P* values for all figures are provided as a source data file. Source data are provided with this paper.

## Code availability
Scripts for peak to gene annotation can be found at https://github.com/sklasfeld/ChIP_Annotation.

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

## Acknowledgements
We thank undergraduate students Gabriela M. Blandino, Xindi Chen and Zubaida Salman and Dietrich James Nigh for help with the experiments, Dr. John D. Wagner for input on the manuscript and the Plant Biology group as well as Wagner lab members for feedback on this project. This research was funded by National Science Foundation IOS grant 1557529 and 1905062 to D.W.

## Author contributions
D.W. and Y.Z. conceived of the study and Y.Z. conducted the majority of the experiments. S.K. conducted the bioinformatic analyses. N.Y. and C.W.J. identified the optimal stage to study primordium fate regulation by TFL1, conducted initial TFL1 ChIP analyses and mapped the FD-binding sites in LFY. R.J. and Y.Z. constructed and sequenced ChIP-seq libraries, K.G generated the biologically active genomic GFP-TFL1 construct. D.W. wrote the manuscript with the help of Y.Z. and input from all other authors.

## Competing interests
The authors declare no competing interests.
