## [Peer Review File · Nature Communications]

REVIEWER COMMENTS

Reviewer #2 (Remarks to the Author):

The manuscript by Zhu et al. describes a genome-wide comparative study of the binding of floral regulators TERMINAL FLOWER 1 (TFL1) and FD. TFL1 and the related FLOWERING LOCUS T (FT), which acts as florigen in the model organism Arabidopsis, play antagonistic function in flowering time regulation and both bind the bZIP transcription factor FD, which is described as positive regulator of flowering based on previous genetic analyses. The presented data test the work hypothesis that TFL1 and FT act as alternative partners in a FD transcription regulatory complex in a competitive and antagonistic manner. For functional genetics-type experiments, the authors focus on the regulation of the floral regulator LEAFY (LFY), an unsuspected direct downstream target of FD, TFL1 and FT. Furthermore, analysis of genome-wide data suggest that the functional trio is implicated in the regulation of gene networks that impact inflorescence development and branching after the reproductive transition.

I have two major concerns, one experimental, one more on the general style of the manuscript. Furthermore, a list of minor points that can be addressed by smaller rewrites are listed below.

1.) The manuscript contains a plenitude of data on carefully designed experiments, the methods, including the bioinformatics pipelines are well described. Given the overall careful and thorough experimental plan, it is even more regrettable that many experiments involving transgenic (reporter and functional) lines seem to be based on single transformants. Given the high variation that can be observed between transgenic lines in expression level and expression pattern, it would be necessary to establish that the selected lines reflect median pattern and expression level within a set of independent lines. This point concerns in particular LFY reporters shown in Figure 2, the comparison between gLFY and gLFYm3 lines shown in Figure 3, it is less problematic for the chemically inducible lines shown in Figure 4.

a. For example Figure 2b shows comparative binding in randomly integrated transgenic lines, which were repeated three times, but it would be appropriate to test biological variation in a set of lines for each construct.

b. Expression pattern of pLFYi2GUS and gLFYGUS as well as their mutant are confirmatory, but one would like to know that these lines are highly representative.

c. Figure 3c, the text states that full complementation is common to 15 gLFY lines but not gLFYm3 lines, the quantitative data show 15 plants. Are these 15 plants of one line (not o.k.) or the actual data from 15 independent lines (as should be)?

2.) The manuscript is written in an extremely concise style, probably to fit a letter format, which not necessarily be imposed by Nature Communications. Although the text is comprehensible for the expert, it remains cryptic for a less informed reader. I strongly suggest to expand the text by including more background information, this would also allow to dig some rather nice elements of the paper out of the Supplemental File (see list below, but that list may not be extensive).

Minor points:

1.) Abstract and introduction: Florigen family of transcriptional regulators is an unconventional shortcut as a majority of the FT-related proteins do not act as florigen (or anti-florigen). Furthermore, florigen is a concept of a mobile signal/hormone. Although FT acts as florigen, it may not be the only florigen in plants. I recommend the use of FT/TFL1 related protein family that act as florigen or anti-florigens in many species. The names are Arabidopsis-specific, it may make sense to move "we use Arabidopsis as model" up one sentence.

2.) "Biological active" gTFL1-GFP line, see comment above. Although the transgenic line differs from the mutant, it is not compared to the WT, thus no full, quantitative complementation analysis is provided. Furthermore, data are from a single line. The same holds true for gFD-GUS, but this a published line and the full complementation data may have been included in the original report.

3.) Figure 1b shows two numbers in the overlap, not sure what they are, one number expected.

4.) Figure 1c: Are the panels co-sorted to the left panel or each sorted independently from strongest to weakest?

- 5.) Figure 1c: Unclear what is shown in the right panel, is it a "GFP-FD-ChIP"? why does the summary plot state "FD-bound?"
- 6.) If Figure 1c shows a GFP ChIP, isn't that redundant to 1e?
- 7.) Figure 2b: it is unclear how the ChIP primer distinguishes the transgene from the endogenous site.
- 8.) Legend Figure 2c: Bottpm
- 9.) Why does a p4kbFT:amiRFT repress FT specifically during the reproductive phase?
- 10.) "Pharmacological upregulation" of FT: One of the examples that the condensed writing is highly problematic. The word does not fit, this is a chemically induced transcription-based reporter system, the term pharmacological would imply that there are chemicals around that directly impact FT function. Needs a few more words on the system and context.
- 11.) Description of Figure 4 in the text is extremely condensed, e.g. "As observed for TFL1 (Fig. 2b), LFY exon two was sufficient and the bZIP sites in it necessary for FT-HAER recruitment (Fig. 4d)." Sufficient for what?

Reviewer #3 (Remarks to the Author):

The work described in the manuscript by Zhu et al. aims to better understand the mechanism of action of TFL1 and FT, two related central regulators of flowering, which control this process in an antagonistic way, with FT acting as an activator of floral genes, and TFL1 as repressor, florigen and antiflorigen, respectively. FT and TFL1, PEBP proteins, lack ability to bind DNA and evidences indicate that they act as transcriptional co-regulators by forming complexes with the bZIP transcription factor FD. FT and TFL1 are highly similar proteins and changes in very few amino acids can make TFL1 to behave as FT and vice versa. This led to a rather accepted model (Ahn et al, 2006) proposing that the basis of the antagonism between the two proteins is that FT and TFL1 compete for binding to FD to form activator or repressor complexes that regulate the same floral genes.

Florigens, FT and TFL1, are conserved proteins among all plants, with a central role in the control of flowering and plant architecture, but their mechanism of action is still not well understood. Therefore, the question posed by the authors, how the florigens modulate plant form - what are the downstream processes they set in motion and what is the molecular basis for their antagonism- , is clearly important for plant developmental biology and plant biology. The manuscript presents a large amount of very sound data, using Arabidopsis as model, aiming to answer that question.

To my view, the major claims of the manuscript, as presented by the authors, are: 1) that TFL1 is recruited to thousands of loci by the bZIP transcription factor FD and that direct TFL1-FD target genes not only identify this complex as repressor of master regulators of flowering, but also implicate TFL1-FD in repression of endogenous signalling pathways, such as sugar and hormones; 2) that the master regulator of floral fate, LFY , is a target under dual opposite regulation by TFL1 and FT and that FT activation of LFY expression is critical to promote floral fate; 3) that the antagonism between FT and TFL1 relies on competition for chromatin-bound FD at shared target loci; and 4) that TFL1 does not merely prevent access of the FT co-activator to the chromatin but is an active repressor.

Regarding importance of the claims, genome-wide identification of TFL1 targets (claim 1) is important, as no clear demonstration that TFL1 does directly regulate transcription was available till very recently, and because it indicates other roles of TFL1 in the regulation of development in addition to flowering. Demonstration that LFY is a target under dual opposite regulation by TFL1 and FT (claim 2) is also very important, and novel, because it is the first clear demonstration that, in fact, FT and TFL1 can compete for binding FD at shared target loci (as suggested in previously

proposed models) and indicates that this mechanism would be the basis of their antagonistic roles (claim 3). Finally, that TFL1 acts as an active repressor (claim 4) is also important for understanding of mechanism of action of florigens.

More detailed comments on particular points of the manuscript follow:

Major points:

1- The genome-wide analysis of TFL1 and FD in this manuscript, claim 1, is very well done and allowed identification of thousands of TFL1-FD target loci. Novelty of this claim is compromised, as a very similar study with the same type of analysis of TFL1 (and FD) targets has been recently published in *Plant Physiology* (Goretti et al., 2020). Nevertheless, the data from this manuscript, of very high quality, expands the data in Goretti et al (2020). As the paper by Goretti et al. (2020), not discussed in this manuscript, contains quite similar data, largely overlapping with these in this manuscript, I think that the results of that paper should be appropriately discussed in this manuscript.

2- The first part of claim 2, that LFY is a target under dual opposite regulation by TFL1 and FT and that FT, I find it important, very well supported and convincing. My only comment is to this first part is that I think that it would be appropriate that some of the previous papers where the model of competition between FT and TFL1 for FD had been previously proposed (as for instance Ahn et al., 2006, and Perilleux et al 2019) are discussed and cited.

3- However, relating the second part of claim 2, that FT promotes flower formation via LFY so that FT activation of LFY expression is critical to promote floral fate, there are some results/questions that I do not fully understand and I think need clarification.

3.1. Pag 5, L109-120 & Fig 2c: Mutation of the bZIP binding sites of LFY exon 2 in LFY-GUS reporters, caused ectopic LFY-GUS expression in the centre of the inflorescence apex while strongly reduced reporter expression in flower primordia (which was considered as suggestion of LFY upregulation by FT). If FT is critical for activation of LFY expression (the main activator?), and these reporters cannot respond to FT, how would LFY expression be activated in the inflorescence meristem?

3.2. Pag 5, L109-120 & Fig 2c: The expression of these LFY-GUS reporters, with mutations in LFY exon 2, seems in contradiction with the data on the LFY promoter by Blázquez et al. (1997), who showed that 2.3 kb of only the 5' of the LFY gene are apparently sufficient to confer the correct LFY expression pattern to a GUS reporter. I think that this apparent contradiction should be discussed.

3.3. Pag 6, L137-149 & Fig 3c,d. (Importance of the bZIP binding sites for inflorescence architecture) Do the lfy plants transformed with the gGLFYm3 construct (genomic LFY with bZIP binding sites mutated) have wild-type flowers or flowers with lfy-like defects? (I think that this should be better illustrated in Fig 3 with close-ups; besides, the Figure legend should say what the arrowheads mark). I think this is a relevant question because if such plants, which due to the bZIP mutations, should "...mimic combined loss of TFL1 and FT activity..", had lfy flowers, then they would be very different to the ft tfl1 mutant, which have wild-type flowers. In addition, the gGLFYm3 plants are described not to have terminal flowers (L146-149). This is different to what occurs in ft tfl1 mutant (Ruiz-García et al, 1997), which have terminal flowers. (And this is in contrast to what it is said in the manuscript, that ft tfl1 mutant lack terminal flowers - referring to Shannon and Meeks-Wagner 1991, where the ft tfl1 mutant is not described). I think that this needs an explanation in the context of the importance of FT activation of LFY expression.

3.4. Finally, ft and ft tfl1 mutants have wild-type flowers. If regulation by FT-FD (through the bZIP

binding sites in the second exon) is that critical to activate LFY expression, shouldn't these mutants have flowers exhibiting some floral defects similar to those of the *lfy* mutants?

4. That FT and TFL1 compete for chromatin-bound FD, not only at the LFY gene but also at other shared target loci, being that the basis of the antagonism between FT and TFL1 (claim 3), is a very important claim, which points the FT-TFL1 competition a general model for the transcriptional regulation by these proteins. The manuscript shows results such as that FRP, which induces FT, induces expression of number of floral regulatory genes that are directly repressed by TFL1-FD, in an FT dependent manner (Supp Fig 12), and that TFL1 occupancy in these loci decreases with FT induction by FRD (Fig. 4f). These data strongly suggests that FT induction upregulates these floral genes by binding to these loci and displacing TFL1 from them. However, even representing a strong indication, I think that these results do not discard the possibility that the FT effect on these loci could be indirect. I believe that experimental results showing direct FT binding to these loci would be required.

5) That TFL1 is an active repressor and does not merely prevent access of the FT co-activator to the chromatin (claim 4), is concluded from the observation that mutating florigen recruitment sites resulted in LFY de-repression in the TFL1 expression domain. Again, I think that this is very suggestive that TFL1 acts as a repressor, but not conclusive, as other alternative explanations, maybe less likely, cannot be discarded. For instance, FD might act a repressor which turns into an activator when forming a complex with FT. Therefore, in my opinion, this claim should be tone down.

6- In the same line, I think that something that would have really meant a major advance for the understanding of how the florigens work would have been to provide some evidences on what is the basis for why FD complexes with very similar proteins, FT and TFL1, work as activator or repressor. Are FT/TFL1 sufficient for these activation/repression activities? Do they have to differentially recruit additional components?

Minor comments:

1. The manuscript is very dense, presenting a lot of experiments and data. Possibly due to that, to my view, quite a few of the experiments are insufficiently explained, to the point that it is not easy to understand them when reading the text. I would suggest that this is revised along the manuscript and, when needed, that explanation on the experiments is included/expanded where required and in the text. A couple of examples are mentioned below.

2. Pag 5 L122, Fig 3A. amiRNA for FT depletion only in the reproductive phase. I think that for easier understanding, how this this amiRNA works, its rational, should be briefly explained in the text (now is explained only in Supp Figure 7).

3. Supp Fig 12. The experiment presented in this figure would need to be explained better in the text and/or the figure legend.

4. Supp Fig 4a-f. I don't think that *lfy* loss-of-function phenocopies TFL1 gain-of-function. It is correct that both *lfy* mutant and 35S:TFL1 plants produce more branches than the wild type (though, obviously, depending on the 35S:TFL1 line which is analysed). However, there are at least two significant differences: 1) 35S:TFL1 plants produce a much higher number of rosette leaves than the wild-type, while *lfy* mutant plants produce the same number of rosette leaves than the wild type and 2) *lfy* mutants never produce normal flowers but their flowers essentially never form petals or stamens while, in contrast, 35S:TFL1 get to produce essentially normal, fertile, flowers

5. Pages 6-7 & Fig 4: There is a discordance in the naming of the Fig 4 panels between the text and the figure.

6. Page 12, References. There is a mistake with reference 18. It should be Kaneko-Suzuki M. et al. rather than Okushita-Terekawa.

References:

Ahn, J. H., Miller, D., Winter, V. J., Banfield, M. J., Lee, J. H., Yoo, S. Y., et al. (2006). A divergent external loop confers antagonistic activity on floral regulators FT and TFL1. *The EMBO Journal*, 25(3), 605–614

Blazquez, M. A., Soowal, L. N., Lee, I., & Weigel, D. (1997). LEAFY expression and flower initiation in *Arabidopsis*. *Development*, 124(19), 3835–3844.

Goretti, D., Silvestre, M., Collani, S., Langenecker, T., Méndez, C., Madueno, F., & Schmid, M. (2020). TERMINAL FLOWER1 Functions as a Mobile Transcriptional Cofactor in the Shoot Apical Meristem. *Plant Physiology*, 182(4), 2081–2095. <http://doi.org/10.1104/pp.19.00867>

Périlleux, C., Bouché, F., Randoux, M., & Orman-Ligeza, B. (2019). Turning Meristems into Fortresses. *Trends in Plant Science*, 1–12. <http://doi.org/10.1016/j.tplants.2019.02.004>

Ruiz-García, L., Madueno, F., Wilkinson, M., Haughn, G., Salinas, J., & Martinez-Zapater, J. M. (1997). Different roles of flowering-time genes in the activation of floral initiation genes in *Arabidopsis*. *The Plant Cell*, 9(11), 1921–1934. <http://doi.org/10.1105/tpc.9.11.1921>

Reviewer #2 (Remarks to the Author):

The manuscript by Zhu et al. describes a genome-wide comparative study of the binding of floral regulators TERMINAL FLOWER 1 (TFL1) and FD. TFL1 and the related FLOWERING LOCUS T (FT), which acts as florigen in the model organism Arabidopsis, play antagonistic function in flowering time regulation and both bind the bZIP transcription factor FD, which is described as positive regulator of flowering based on previous genetic analyses. The presented data test the work hypothesis that TFL1 and FT act as alternative partners in a FD transcription regulatory complex in a competitive and antagonistic manner. For functional genetics-type experiments, the authors focus on the regulation of the floral regulator LEAFY (LFY), an unsuspected direct downstream target of FD, TFL1 and FT. Furthermore, analysis of genome-wide data suggest that the functional trio is implicated in the regulation of gene networks that impact inflorescence development and branching after the reproductive transition. I have two major concerns, one experimental, one more on the general style of the manuscript. Furthermore, a list of minor points that can be addressed by smaller rewrites are listed below.

Reviewer #2

1.) The manuscript contains a plenitude of data on carefully designed experiments, the methods, including the bioinformatics pipelines are well described. Given the overall careful and thorough experimental plan, it is even more regrettable that many experiments involving transgenic (reporter and functional) lines seem to be based on single transformants. Given the high variation that can be observed between transgenic lines in expression level and expression pattern, it would be necessary to establish that the selected lines reflect median pattern and expression level within a set of independent lines. This point concerns in particular LFY reporters shown in Figure 2, the comparison between gLFY and gLFYm3 lines shown in Figure 3, it is less problematic for the chemically inducible lines shown in Figure 4.

Reviewer #2

a. For example Figure 2b shows comparative binding in randomly integrated transgenic lines, which were repeated three times, but it would be appropriate to test biological variation in a set of lines for each construct.

Authors: We apologize that we did not explain this better. To generate the double transgenic lines used in Fig 2b (now Fig. 2a) and Fig 4e, we transformed gTFL1-GFP *tfl1-1* and 35S:FT-HA^{ER} with binary vectors containing only LFYe2 or LFYe2m3. We conducted ChIP q-PCR on random progeny pools of >50 T₁ transformants. As in a previous publication (Xiao et al. Nature Genetics 2017), we used this approach to minimize possible position effects on the transgenes.

Reviewer #2

b. Expression pattern of pLFYi2GUS and gLFYGUS as well as their mutant are confirmatory, but one would like to know that these lines are highly representative.

Authors: We fully agree. For the above-mentioned lines, GUS staining was performed in independent T₂ transformants and several representative lines were chosen for further analysis. We have added pictures of these additional lines in Supplementary figure 6 panel d. GUS staining was performed under identical experimental settings as well developmental stages. We have added this information to the methods.

Reviewer #2

c. Figure 3c, the text states that full complementation is common to 15 gLFY lines but not gLFYm3 lines, the quantitative data show 15 plants. Are these 15 plants of one line (not o.k.) or the actual data from 15 independent lines (as should be)?

Authors: Thank you for pointing out that this was not clear. The data in Fig. 3 is from 15 independent transgenic lines each for gLFY and for gLFYm3. For gLFY lfy-1 we in generated 25 plants in total, 24 of which rescued. The data is from 15 randomly chosen rescuing plants. For gLFYm3 we generated 15 independent lines and show the data for all of these. We now state that these are independent lines in the figure legend. We have also included a photo of one additional independent transformant per construct in Supplementary Fig. 10c.

Reviewer #2

2.) The manuscript is written in an extremely concise style, probably to fit a letter format, which not necessarily be imposed by Nature Communications. Although the text is comprehensible for the expert, it remains cryptic for a less informed reader. I strongly suggest to expand the text by including more background information, this would also allow to dig some rather nice elements of the paper out of the Supplemental File (see list below, but that list may not be extensive).

Authors: We agree with the reviewer and have revised the text accordingly.

Reviewer #2

Minor points:

1.) Abstract and introduction: Florigen family of transcriptional regulators is an unconventional shortcut as a majority of the FT-related proteins do not act as florigen (or anti-florigen). Furthermore, florigen is a concept of a mobile signal/hormone. Although FT acts as florigen, it may not be the only florigen in plants. I recommend the use of FT/TFL1 related protein family that act as florigen or anti-florigens in many species. The names are Arabidopsis-specific, it may make sense to move “we use Arabidopsis as model” up one sentence.

Authors: We have replaced the term ‘florigen’ with FT/TFL1 and FT/TFL1-like gene function or PEBP family member throughout the manuscript.

Reviewer #2

2.) “Biological active” gTFL1-GFP line, see comment above. Although the transgenic line differs from the mutant, it is not compared to the WT, thus no full, quantitative complementation analysis is provided. Furthermore, data are from a single line. The same holds true for gFD-GUS, but this a published line and the full complementation data may have been included in the original report.

Authors: We thank the reviewer for these suggestions. We have included a wild-type control to allow comparison with the rescue line for both gTFL1 and gFD-GUS. We also include images of additional gTFL1-GFP *tfl1-1* lines (see revised Supplementary Fig. 1).

Reviewer #2

3.) Figure 1b shows two numbers in the overlap, not sure what they are, one number expected.

Authors: ChIP peaks in different datasets generally do not entirely overlap. This means that multiple ChIP peaks from ChIP A can overlap with a single ChIP peak in ChIP B and vice-versa. To address this, we display the number of peaks in ChIP B that ChIP A overlaps with and the number of peaks in ChIP A that ChIP B overlaps with. Here 2372 TFL1 peaks (purple) overlap with FD ChIPseq peaks and 2349 FD peaks (orange) overlap with the TFL1 ChIPseq peaks.

Reviewer #2

4.) Figure 1c: Are the panels co-sorted to the left panel or each sorted independently from strongest to weakest?

Authors: The heatmaps were centred on the TFL1 peak summits and ranked (co-sorted) from highest to lowest TFL1 ChIP-seq signal. We have added this information to the figure legend.

Reviewer #2

5.) Figure 1c: Unclear what is shown in the right panel, is it a “GFP-FD-ChIP”? why does the summary plot state “FD-bound?”

Authors: We decided to remove these plots because of point 6. below

Reviewer #2

6.) If Figure 1c shows a GFP ChIP, isn't that redundant to 1e?

Authors: We agree and have added the FD ChIP-seq signal under TFL1 peaks to this plot (formerly Fig. 1e), which already showed the TFL1 ChIP-seq signal in the wild type and fd mutants.

Reviewer #2

7.) Figure 2b: it is unclear how the ChIP primer distinguishes the transgene from the endogenous site.

Authors: This information was missing from the figure legend but is now added. P1 and P2 are two amplicons that each span the LFYe2 or LFYe2m3 region and the vector sequence (grey line). The reverse primer of P1 and the forward primer of P2 are specific to LFY exon 2. The forward primer of P1 and the reverse primer of P2 recognize the sequence of the vector at the flank of transgene LFYe2 or LFYe2m3.

Reviewer #2

8.) Legend Figure 2c: Bottpm

Authors: We corrected this typo.

Reviewer #2

9.) Why does a p4kbFT:amiRFT repress FT specifically during the reproductive phase?

Authors: Reference 79 had shown that the minimal 4 kb FT promoter is active in some parts of the leaves, in the stem and below the shoot apical meristem from day 12 onward in long-day grown plants. The authors further demonstrate that p4kbFT:FT does not rescue the late flowering phenotype of *ft* mutants. We linked p4kbFT to a previously described artificial microRNA specific to FT (Schwab et al. PC) to reduce FT accumulation specifically during the reproductive phase of development.

Reviewer #2

10.) "Pharmacological upregulation" of FT: One of the examples that the condensed writing is highly problematic. The word does not fit, this is a chemically induced transcription-based reporter system, the term pharmacological would imply that there are chemicals around that directly impact FT function. Needs a few more words on the system and context.

Authors: We modified the text to address this concern.

Reviewer #2

11.) Description of Figure 4 in the text is extremely condensed, e.g. "As observed for TFL1 (Fig. 2b), LFY exon two was sufficient and the bZIP sites in it necessary for FT-HAER recruitment (Fig. 4d)." Sufficient for what?

Authors: We modified the text to address this concern.

Reviewer #3 (Remarks to the Author):

The work described in the manuscript by Zhu et al. aims to better understand the mechanism of action of TFL1 and FT, two related central regulators of flowering, which control this process in an antagonistic way, with FT acting as an activator of floral genes, and TFL1 as repressor, florigen and antiflorigen, respectively. FT and TFL1, PEBP proteins, lack ability to bind DNA and evidences indicate that they act as transcriptional co-regulators by forming complexes with the bZIP transcription factor FD. FT and TFL1 are highly similar proteins and changes in very few amino acids can make TFL1 to behave as FT and vice versa. This led to a rather accepted model (Ahn et al, 2006) proposing that the basis of the antagonism between the two proteins is that FT and TFL1 compete for binding to FD to form activator or repressor complexes that regulate the same floral genes.

Florigens, FT and TFL1, are conserved proteins among all plants, with a central role in the control of flowering and plant architecture, but their mechanism of action is still not well understood. Therefore, the question posed by the authors, how the florigens modulate plant form - what are the downstream processes they set in motion and what is the molecular basis for their antagonism- , is clearly important for plant developmental biology and plant biology. The manuscript presents a large amount of very sound data, using Arabidopsis as model, aiming to answer that question.

To my view, the major claims of the manuscript, as presented by the authors, are: 1) that TFL1 is recruited to thousands of loci by the bZIP transcription factor FD and that direct TFL1-FD target genes not only identify this complex as repressor of master regulators of flowering, but also implicate TFL1-FD in repression of endogenous signalling pathways, such as sugar and hormones; 2) that the master regulator of floral fate, LFY , is a target under dual opposite regulation by TFL1 and FT and that FT activation of LFY expression is critical to promote floral fate; 3) that the antagonism between FT and TFL1 relies on competition for chromatin-bound FD at shared target loci; and 4) that TFL1 does not merely prevent access of the FT co-activator to the chromatin but is an active repressor.

Regarding importance of the claims, genome-wide identification of TFL1 targets (claim 1) is important, as no clear demonstration that TFL1 does directly regulate transcription was available till very recently, and because it indicates other roles of TFL1 in the regulation of development in addition to flowering. Demonstration that LFY is a target under dual opposite regulation by TFL1 and FT (claim 2) is also very important, and novel, because it is the first clear demonstration that, in fact, FT and TFL1 can compete for binding FD at shared target loci (as suggested in previously proposed models) and indicates that this mechanism would be the basis of their antagonistic roles (claim 3). Finally, that TFL1 acts as an active repressor (claim 4) is also important for understanding of mechanism of action of florigens.

More detailed comments on particular points of the manuscript follow:

Major points:

Regarding importance of the claims, genome-wide identification of TFL1 targets (claim 1) is important, as no clear demonstration that TFL1 does directly regulate transcription was available till very recently, and because it indicates other roles of TFL1 in the regulation of development in addition to flowering.

Reviewer #3

1- The genome-wide analysis of TFL1 and FD in this manuscript, claim 1, is very well done and allowed identification of thousands of TFL1-FD target loci. Novelty of this claim is compromised, as a very similar study with the same type of analysis of TFL1 (and FD) targets has been recently published in Plant Physiology (Goretti et al., 2020). Nevertheless, the data from this manuscript, of very high quality, expands the data in Goretti et al (2020). As the paper by Goretti et al. (2020), not discussed in this manuscript, contains quite similar data, largely overlapping with these in this manuscript, I think that the results of that paper should be appropriately discussed in this manuscript.

Authors:

The Goretti et al. paper and submission of this manuscript coincided. We agree Goretti et al. should be discussed and now do so. As we had done for the Collani et al. data in the initial submission, we now include the Goretti et al. data in the Pearson correlation co-efficient analysis of ChIP-seq replicates in the updated Supplementary Fig. 3a. We also identified peaks and peak associated genes for Goretti (TFL1 ChIPseq) and Collani (FD ChIPseq) using our computational pipelines. Despite having been conducted in different conditions (long day (Collani and Goretti) versus short day (our data)), these analyses reveal excellent overlap between peak associated genes in all four datasets and furthermore confirms the list of 604 TFL1/FD bound and repressed genes we identify in the current study (see new Supplementary Fig. 3b).

Reviewer #3

2) that the master regulator of floral fate, LFY, is a target under dual opposite regulation by TFL1 and FT and that FT activation of LFY expression is critical to promote floral fate;

2- The first part of claim 2, that LFY is a target under dual opposite regulation by TFL1 and FT and that FT, I find it important, very well supported and convincing. My only comment is to this first part is that I think that it would be appropriate that some of the previous papers where the model of competition between FT and TFL1 for FD had been previously proposed (as for instance Ahn et al., 2006, and Perilleux et al 2019) are discussed and cited.

Authors: We that agree that this model for the antagonism between has been proposed and think that the best in vivo evidence to date for it is that 35S:TFL1-VP16 showed early flowering while 35S:TFL1-SRX, like 35S:TFL1, delayed it (Hanano and Goto

2011). We did already cite the excellent review from Perilleux, and have now added a section to the introduction that covers the interconvertibility of the FT and TFL1 proteins described by Ahn et al. 2006 and additional publications, which also support this model.

Reviewer #3

3) that the antagonism between FT and TFL1 relies on competition for chromatin-bound FD at shared target loci;

3- However, relating the second part of claim 2, that FT promotes flower formation via LFY so that FT activation of LFY expression is critical to promote floral fate, there are some results/questions that I do not fully understand and I think need clarification.

3.1. Pag 5, L109-120 & Fig 2c: Mutation of the bZIP binding sites of LFY exon 2 in LFY-GUS reporters, caused ectopic LFY-GUS expression in the centre of the inflorescence apex while strongly reduced reporter expression in flower primordia (which was considered as suggestion of LFY upregulation by FT). If FT is critical for activation of LFY expression (the main activator?), and these reporters cannot respond to FT, how would LFY expression be activated in the inflorescence meristem?

Authors:

We apologize if this was unclear; the ectopic LFY accumulation in the centre of the inflorescence meristem is due to other factors, not FT/FD. Since we mutated the FD binding sites in LFY in the pLFYi2m3:GUS and gLFYm3:GUS, FT can no longer activate these reporters (Fig. 3b, Supplementary Fig. 9e). The identity of the transcriptional activators responsible for the ectopic LFY expression is not known. SOC1 or SPL proteins (Lee et al. PJ 2008, Hyun et al. Dev Cell 2016, Yamaguchi et al. Dev Cell 2009) are plausible candidates as both sets of transcription factors are expressed in the inflorescence shoot apex, and can bind to and activate the *LFY* locus. We have revised the main text and the discussion to clarify this.

Reviewer #3

3.2. Pag 5, L109-120 & Fig 2c: The expression of these LFY-GUS reporters, with mutations in LFY exon 2, seems in contradiction with the data on the LFY promoter by Blázquez et al. (1997), who showed that 2.3 kb of only the 5' of the LFY gene are apparently sufficient to confer the correct LFY expression pattern to a GUS reporter. I think that this apparent contradiction should be discussed.

Authors: Our genomic LFY (gLFY) reporters include the 2.3kb promoter used by Blázquez et al. (Dev 1997) plus three exons and two introns (gLFY:GUS and gGLFY; Fig 2c and Fig3c) or exons 1 and 2 plus both introns (pLFYi2). Of note, the genomic LFY reporters and pLFYi2 more closely approximate the endogenous locus with respect to the regulatory regions that may act on *LFY*. They furthermore exhibit expression patterns that closely parallel that of endogenous *LFY* and rescue the meristem identity and floral homeotic defects of *lfy-2*. pLFY for the most part directs reporter expression similar to endogenous *LFY*, but shows stronger expression in vegetative tissues than endogenous *LFY*. That LFY expression so dramatically reduced in pLFYi2m3:GUS,

gLFYm3:GUS, gGLFYm3, all of which include the 2.3kb LFY promoter, suggests that the genic *LFY* region contains additional regulatory elements. These unknown regulatory elements likely prevent pLFY activity in the context of the genomic constructs, in which the bZIP binding sites are mutated. We revised the text to include these considerations.

Reviewer #3

3.3. Pag 6, L137-149 & Fig 3c,d. (Importance of the bZIP binding sites for inflorescence architecture) Do the *lfy* plants transformed with the gGLFYm3 construct (genomic LFY with bZIP binding sites mutated) have wild-type flowers or flowers with *lfy*-like defects? (I think that this should be better illustrated in Fig 3 with close-ups; besides, the Figure legend should say what the arrowheads mark). I think this is a relevant question because if such plants, which due to the bZIP mutations, should "...mimic combined loss of TFL1 and FT activity..", had *lfy* flowers, then they would be very different to the *ft tfl1* mutant, which have wild-type flowers. In addition, the gGLFYm3 plants are described not to have terminal flowers (L146-149). This is different to what occurs in *ft tfl1* mutant (Ruiz-García et al, 1997), which have terminal flowers. (And this is in contrast to what it is said in the manuscript, that *ft tfl1* mutant lack terminal flowers - referring to Shannon and Meeks-Wagner 1991, where the *ft tfl1* mutant is not described). I think that this needs an explanation in the context of the importance of FT activation of LFY expression.

We have subdivided the reviewer queries into three sections addressed below:

Reviewer #3

3.3 A. The reviewer asks whether gGLFYm3 *lfy* plants have wild-type flowers or flowers with patterning defects, given that mutants in PEBP proteins do not have such defects?

Authors:

gGLFYm3 *lfy* flowers have patterning defects. By contrast, *ft tsf* double mutants form many more branches, but give rise to normal flowers. However, TFL1 overexpression also results in flower patterning defects (Ho et al. PC 2014). Why gGLFYm3 *lfy* and gain-of-function TFL1 differ from loss of activating PEBP protein function with respect to flower patterning is not understood. While this may be an interesting topic for future investigation, we suggest that the possible role of PEBP proteins in flower patterning is beyond the focus of the current manuscript.

Reviewer #3

3.3 B. The Figure legend should say what the arrowheads mark.

Authors:

We thank the reviewer for pointing out this oversight. The white arrowheads indicates the branches on the main stem. This information was added to the figure legend.

Reviewer #3

3.3 C. In addition, the gGLFYm3 plants are described not to have terminal flowers (L146-149). This is different to what occurs in *ft tfl1* mutant (Ruiz-García et al, 1997), which have terminal flowers. (And this is in contrast to what it is said in the manuscript, that *ft tfl1* mutant lack terminal flowers - referring to Shannon and Meeks-Wagner 1991, where the *ft tfl1* mutant is not described).

Authors:

Indeed, the terminal flower phenotype of *tfl1* is partially attenuated in *tfl1* mutants grown in short day conditions, which form >40 normal flowers before eventually terminating (Shannon et al., PC 1991). Likewise, *ft-1 tfl1-2* still terminates (Ruiz Gracia et al. PC 1997). More recently Lee et al. PJ 2019 provided compelling evidence for a pivotal role for FT and its closest homolog TWIN SISTER OF FT (TSF) in terminal flower formation. Removal of both FT and TSF entirely abolishes the terminal flower phenotype of *tfl1* (Lee et al. PJ 2019). We have revised the text accordingly.

Reviewer #3

4. That FT and TFL1 compete for chromatin-bound FD, not only at the LFY gene but also at other shared target loci, being that the basis of the antagonism between FT and TFL1 (claim 3), is a very important claim, which points the FT-TFL1 competition a general model for the transcriptional regulation by these proteins. The manuscript shows results such as that FRP, which induces FT, induces expression of number of floral regulatory genes that are directly repressed by TFL1-FD, in an FT dependent manner (Supp Fig 12), and that TFL1 occupancy in these loci decreases with FT induction by FRD (Fig. 4f). These data strongly suggests that FT induction upregulates these floral genes by binding to these loci and displacing TFL1 from them. However, even representing a strong indication, I think that these results do not discard the possibility that the FT effect on these loci could be indirect. I believe that experimental results showing direct FT binding to these loci would be required.

Authors:

We agree with the reviewer's concern and have now conducted FT-HA^{ER} ChIP-qPCR. FT is indeed recruited to the TFL1 and FD bound sites at all flowering loci tested. The new data is available in Fig. 4g.

5) That TFL1 is an active repressor and does not merely prevent access of the FT co-activator to the chromatin (claim 4), is concluded from the observation that mutating florigen recruitment sites resulted in LFY de-repression in the TFL1 expression domain. Again, I think that this is very suggestive that TFL1 acts as a repressor, but not conclusive, as other alternative explanations, maybe less likely, cannot be discarded. For instance, FD might act a repressor which turns into an activator when forming a complex with FT. Therefore, in my opinion, this claim should be tone down.

Authors:

We agree with this suggestion. We have modified the text to tone this conclusion down.

Reviewer #3

6- In the same line, I think that something that would have really meant a major advance for the understanding of how the florigens work would have been to provide some evidences on what is the basis for why FD complexes with very similar proteins, FT and TFL1, work as activator or repressor. Are FT/TFL1 sufficient for these activation/repression activities? Do they have to differentially recruit additional components?

Authors:

We agree with the reviewer that this is a very important question. However, answering this question will require an entire new investigation and is therefore beyond the scope of the current MS.

Minor comments:

Reviewer #3

1. The manuscript is very dense, presenting a lot of experiments and data. Possibly due to that, to my view, quite a few of the experiments are insufficiently explained, to the point that it is not easy to understand them when reading the text. I would suggest that this is revised along the manuscript and, when needed, that explanation on the experiments is included/expanded where required and in the text. A couple of examples are mentioned below.

2. Pag 5 L122, Fig 3A. amiRNA for FT depletion only in the reproductive phase. I think that for easier understanding, how this this amiRNA works, its rational, should be briefly explained in the text (now is explained only in Supp Figure 7).

Authors:

We have added more information about this experiment in the text.

Reviewer #3

3. Supp Fig 12. The experiment presented in this figure would need to be explained better in the text and/or the figure legend.

Authors:

We have added additional information to explain supplementary Fig. 12

Reviewer #3

4. Supp Fig 4a-f. I don't think that lfy loss-of-function phenocopies TFL1 gain-of-function. It is correct that both lfy mutant and 35S:TFL1 plants produce more branches than the wild type (though, obviously, depending on the 35S:TFL1 line which is analysed). However, there are at least two significant differences: 1) 35S:TFL1 plants produce a much higher number of rosette leaves than the wild-type, while lfy mutant plants produce the same number of rosette leaves than the wild type and 2) lfy mutants

never produce normal flowers but their flowers essentially never form petals or stamens while, in contrast, 35S:TFL1 get to produce essentially normal, fertile, flowers

Authors:

We agreed with this concern and have rephrased the text and Figure legend accordingly.

Reviewer #3

5. Pages 6-7 & Fig 4: There is a discordance in the naming of the Fig 4 panels between the text and the figure.

Authors:

Thank you for pointing out this issue, which we have fixed.

Reviewer #3

6. Page 12, References. There is a mistake with reference 18. It should be Kaneko-Suzuki M. et al. rather than Okushita-Terekawa.

Authors:

Thank you for pointing out this issue, which we have fixed.

REVIEWERS' COMMENTS:

Reviewer #2 (Remarks to the Author):

The authors have addressed all of my previous concerns to my full satisfaction.

Reviewer #3 (Remarks to the Author):

I think that this new version of the manuscript by Zhu et al. introduces changes that provide satisfactory answers to most of the concerns that I raised in my previous review. I think that the new FT-HAER ChIP-qPCR data, which shows binding of FT to several flowering loci also bound by TFL1, is particularly interesting, as it shows with no doubt that the antagonism between FT and TFL1 is based on competition of these proteins for transcriptional regulation of shared target loci, a model that has been around for a long time but had not been proved.

However, I believe that some previous literature has not been fairly treated. There are two cases that stand out, which I already pointed in my previous review (major points 1 and 2), but that I don't think have been properly amended in this new version.

1- TFL1-FD targets, a main claim of the manuscript, have been already reported (Goretti et al., Plant Phys 2020) and this has to be more explicitly described, rather than being cited in a non-informative statement (page 3: "... FT and TFL1 are small mobile proteins, which have been implicated in transcriptional regulation but do not have DNA binding domains 14-18 ...") and in a supplementary figure (where the citation has not even been included). I believe that this has to be amended because otherwise it would appear that this is the first time that this data is being presented, which would be unfair with the previous work.

2- The idea of competition of FT and TFL1 for binding to their targets is also central to this manuscript. The model that FT and TFL1 compete for binding to their targets, as basis for their antagonism, was proposed long ago and has been discussed and cited quite many times. To my knowledge, as such model, it was proposed by Ahn et al. (EMBO J, 2006). Though Ahn et al. 2006 and some other papers where the model is described have been cited in the manuscript, they have been cited in a different context, and thus, the previous knowledge regarding the model has not been described, it has not been put into context, with the cites in the proper sites.

REVIEWERS' COMMENTS:

Reviewer #2 (Remarks to the Author):

The authors have addressed all of my previous concerns to my full satisfaction.

Reviewer #3 (Remarks to the Author):

I think that this new version of the manuscript by Zhu et al. introduces changes that provide satisfactory answers to most of the concerns that I raised in my previous review. I think that the new FT-HAER ChIP-qPCR data, which shows binding of FT to several flowering loci also bound by TFL1, is particularly interesting, as it shows with no doubt that the antagonism between FT and TFL1 is based on competition of these proteins for transcriptional regulation of shared target loci, a model that has been around for a long time but had not been proved.

However, I believe that some previous literature has not been fairly treated. There are two cases that stand out, which I already pointed in my previous review (major points 1 and 2), but that I don't think have been properly amended in this new version.

1- TFL1-FD targets, a main claim of the manuscript, have been already reported (Goretti et al., Plant Phys 2020) and this has to be more explicitly described, rather than being cited in a non-informative statement (page 3: "... FT and TFL1 are small mobile proteins, which have been implicated in transcriptional regulation but do not have DNA binding domains 14-18 ...") and in a supplementary figure (where the citation has not even been included). I believe that this has to be amended because otherwise it would appear that this is the first time that this data is being presented, which would be unfair with the previous work.

We respectfully beg to differ regarding the reviewer's contention that TFL1-FD [regulated] targets were already reported by Goretti 2020. Using the same criteria as we apply in our study for identification of TFL1 and FD co-bound and repressed genes, Goretti uncovers 14 TFL1-FD targets, none of which are known flowering or meristem identity regulators. These 14 genes are arrived at by overlapping the 562 genes identified by Goretti as significantly downregulated in response to TFL1-GR (Goretti Supplemental data 2) with the 385 genes co-bound by TFL1 and FD (Goretti Supplemental data 4). By contrast, our study identified 604 immediate early TFL1-FD repressed targets, including key flowering time and meristem identity regulators. To understand this apparent discrepancy, we reanalyzed the recently published TFL1 and FD ChIPseq datasets (Collani 2019 and Goretti 2020). As shown in Fig. S3b, 43% of the 604 immediate early TFL1 and FD regulated genes we identified in our study were bound by both pTFL1:TFL1 (Goretti) and pFD:FD (Collani), while 82% of the 604 TFL1-FD bound and regulated loci are bound by at least one of these recently published datasets. The overlap includes the flowering time and meristem identity genes. We have rephased the statement in the text to better underscore this point. We would also like to point out that the Goretti reference was (and still is) cited in the legend of revised Fig. S3 (Supplemental reference 4).

2- The idea of competition of FT and TFL1 for binding to their targets is also central to this manuscript. The model that FT and TFL1 compete for binding to their targets, as basis for their antagonism, was proposed long ago and has been discussed and cited quite many times. To my knowledge, as such model, it was proposed by Ahn et al. (EMBO J, 2006). Though Ahn et al. 2006 and some other papers where the model is described have been cited in the

manuscript, they have been cited in a different context, and thus, the previous knowledge regarding the model has not been described, it has not been put into context, with the cites in the proper sites.

Thank you for the discussion of the antagonistic roles of Arabidopsis FT and TFL1. We will address this concern in our rebuttal here in two steps:

- 1. The hypothesis that TFL1 and FT might compete a common factor (FD) was arrived at by studies in many laboratories over two decades and cannot be attributed to a single publication. Antagonism between FT and TFL1 was discussed as early as 1999 (Kobayashi et al., Science). This was followed by two papers in 2005 that reveal FT and TFL1 can physically interact with FD, and that the FT gain-of-function phenotype is partially dependent on fd (Wigge 2005 Science, Abe 2005 Science). Hanzawa 2005 showed that TFL1 can be converted into FT and vice versa via single amino-acid changes and Ahn et al. EMBO J 2006 demonstrated that the external loop is important for unique FT and TFL1 activities and suggested (on the basis of the combined studies above) competition for a common interaction partner (FD). Further support came from Taoka 2011, Ho 2014 and Kaneko Suzuki 2018. To our mind, critical in vivo evidence for possible competition at target loci came from Hanano et al 2011.*
- 2. We show that FD and TFL1 compete for binding to shared target loci and that FT recruitment leads to TFL1 loss from target loci, while FD levels remain unchanged. Our findings are based on chromatin immunoprecipitation followed by sequencing (TFL1 ChIPseq, FD ChIPseq and TFL1 fd ChIPseq) and RNAseq in diverse mutants and wild-type backgrounds in response to photoperiod shift combined with in-depth case studies at the LFY and now other flowering target loci. Indeed, the reviewer states above that the new “FT-HAER ChIP-qPCR data, which shows binding of FT to several flowering loci also bound by TFL1, is particularly interesting, as it shows with no doubt that the antagonism between FT and TFL1 is based on competition of these proteins for transcriptional regulation of shared target loci.” There are still remaining questions regarding the antagonism between PEBP proteins and we look forward to delving deeper into this question ourselves and to further studies on this topic by others.*